# The recurrent temporal restricted Boltzmann machine captures neural assembly dynamics in whole-brain activity

Sebastian Quiroz Monnens[†‡], Casper Peters[†], Luuk Willem Hesselink[†],
Kasper Smeets[§], Bernhard Englitz*

Computational Neuroscience Lab, Donders Center for Neuroscience, Radboud University, Nijmegen, Netherlands

*For correspondence:
bernhard.englitz@donders.ru.nl

[†]These authors contributed equally to this work

Present address: [‡]Center for Neurogenomics andCognitive Research, VrijeUniversiteit Amsterdam, Amsterdam, Netherlands; [§]École Polytechnique Fédérale deLausanne, Lausanne, Switzerland

Competing interest: The authors declare that no competing interests exist.

## eLife Assessment

This study introduces a **useful** extension to a recently proposed model of neural assembly activity. The extension was to add recurrent connections to the hidden units of the Restricted Boltzmann Machine. The authors show **solid** evidence that the new model outperforms their earlier model on both a simulated dataset and on whole-brain neural activity from zebrafish.

**Abstract** Animal behaviour alternates between stochastic exploration and goal-directed actions, which are generated by the underlying neural dynamics. Previously, we demonstrated that the compositional Restricted Boltzmann Machine (cRBM) can decompose whole-brain activity of larval zebrafish data at the neural level into a small number (~100-200) of assemblies that can account for the stochasticity of the neural activity (van der Plas et al., eLife, 2023). Here, we advance this representation by extending to a combined stochastic-dynamical representation to account for both aspects using the recurrent temporal RBM (RTRBM) and transfer-learning based on the cRBM estimate. We demonstrate that the functional advantage of the RTRBM is captured in the temporal weights on the hidden units, representing neural assemblies, for both simulated and experimental data. Our results show that the temporal expansion outperforms the stochastic-only cRBM in terms of generalization error and achieves a more accurate representation of the moments in time. Lastly, we demonstrate that we can identify the original time-scale of assembly dynamics by estimating multiple RTRBMs at different temporal resolutions. Together, we propose that RTRBMs are a valuable tool for capturing the combined stochastic and time-predictive dynamics of large-scale data sets.

## Introduction

When large groups of neurons exhibit joint activity, they are often assumed to form a functional unit, referred to as a neural assembly (*Harris, 2005*). Neural assemblies are thought to form elementary computational units that are essential for cognitive functions such as short-term memory, sensorimotor computation, and decision-making (*Harris, 2005*; *Hebb, 1949*; *Gerstein et al., 1989*). Recent advancements in neuroimaging methods now enable us to study the role of these neural assemblies in more detail. For example, a large breakthrough is the introduction of light-sheet microscopy which enables functional recordings of whole-brain volumes, thereby allowing the study of how complex computation emerges in the brain (*Ahrens et al., 2013*). It, however, remains a computational challenge to extract neural activation patterns from such datasets comprising ~100.000 neurons or more.

Recent work leveraged the compositional restricted Boltzmann machine (cRBM) to identify such neural assemblies in large-scale neural data (*van der Plas et al., 2023*). The cRBM is an extension of the restricted Boltzmann machine (RBM) (*Smolensky, 1986*), an undirected graphical model that consists of two layers of random variables, representing the data itself (through a set of visible units) and a lower-dimensional latent representation (through a set of hidden units). The model learns in an unsupervised manner by matching its model distribution to the empirical distribution of the data through maximum likelihood optimization (*Tubiana et al., 2019a*; *Salakhutdinov et al., 2007*). The cRBM extends the classical RBM by adhering to a set of structural conditions (see Materials and methods), pushing it to operate in a state referred to as the compositional phase (*Tubiana and Monasson, 2017*; *Tubiana et al., 2019a*; *Tubiana et al., 2019b*). In the compositional phase, the visible-to-hidden connections in the RBM are sparse, and in our previous work (*van der Plas et al., 2023*) this Research Advance is based on, we show that the associated neural assemblies of the hidden units are localized and span the entire space of the visible units. Furthermore, only a small fraction of the hidden units are active at any point in time, improving model interpretability.

Although the cRBM can accurately reproduce neural statistics and produce a low-dimensional representation of the high-dimensional neural data, this model is limited in capturing only static dependencies and is unable to specifically account for temporal dependencies. Neural activity driving animal behavior is expressed in both stochastic and deterministic states, thus requiring dynamics to be explicitly included to capture most of the variance in the data. To tackle this problem, we here include temporal dependencies directly into the model by applying the recurrent temporal RBM (RTRBM) (*Sutskever et al., 2008*). We utilize a type of transfer learning to retain the sparsity advantages of the cRBM, while the model can additionally account for the deterministic dynamics underlying the neural activity it's trained on.

In short, the RTRBM is a recurrent neural network constructed by chaining multiple RBMs in time (*Mittelman et al., 2014*). Each RBM has a hidden state that is conditioned on the expected hidden state of the RBM at the preceding time-step. While temporal connections are constrained to single time-steps, the recurrency in the model indirectly accounts for multi-time-step dependencies. Previous studies using RTRBMs in other domains have highlighted the value of including such temporal dependencies in extracting spatiotemporal features from high-dimensional data (*Boulanger-Lewandowski et al., 2012*; *Li et al., 2018*; *Zhang et al., 2018*). A more detailed discussion on the RTRBM and its implementation can be found in RTRBM.

Results In this work, we apply the RTRBM to both simulated and real data. First, we show that the model is capable of retrieving the artificial neural assemblies and their temporal connections in a fully simulated networks with only a few hidden populations. We compare the resulting RTRBM with an RBM that is trained on the same data, and shows that it outperforms in terms of generalization error, pairwise moments, and time-shifted pairwise moments. We then use a combined approach of the RTRBM and the cRBM to model the temporal connections of different neural assemblies in whole-brain neuronal zebrafish data through transfer learning by initializing RTRBM weights with their cRBM counterparts. The resulting RTRBM model extends upon the neural assemblies identified by the cRBM models by additionally capturing their temporal dependencies. We demonstrate that this extension improves the reconstructive power in terms of the estimated moments and provides additional temporal information regarding the underlying structure of the brain.

## Results

In this work, we investigated whether the inclusion of temporal dependencies between neural assemblies improve the representation of neuronal activity in the context of simulated data and whole-brain light-sheet recordings from zebrafish larvae ($n = 8$, $40709 \pm 13854$ neurons, *van der Plas et al., 2023*). For this purpose, we first introduce and compare the (RTRBM) to the compositional RBM (*Tubiana et al., 2019a*) and then use two-step transfer learning to arrive at an estimate of the RTRBM that stays in the compositional phase and maintains the locally restricted assemblies identified by the cRBM.

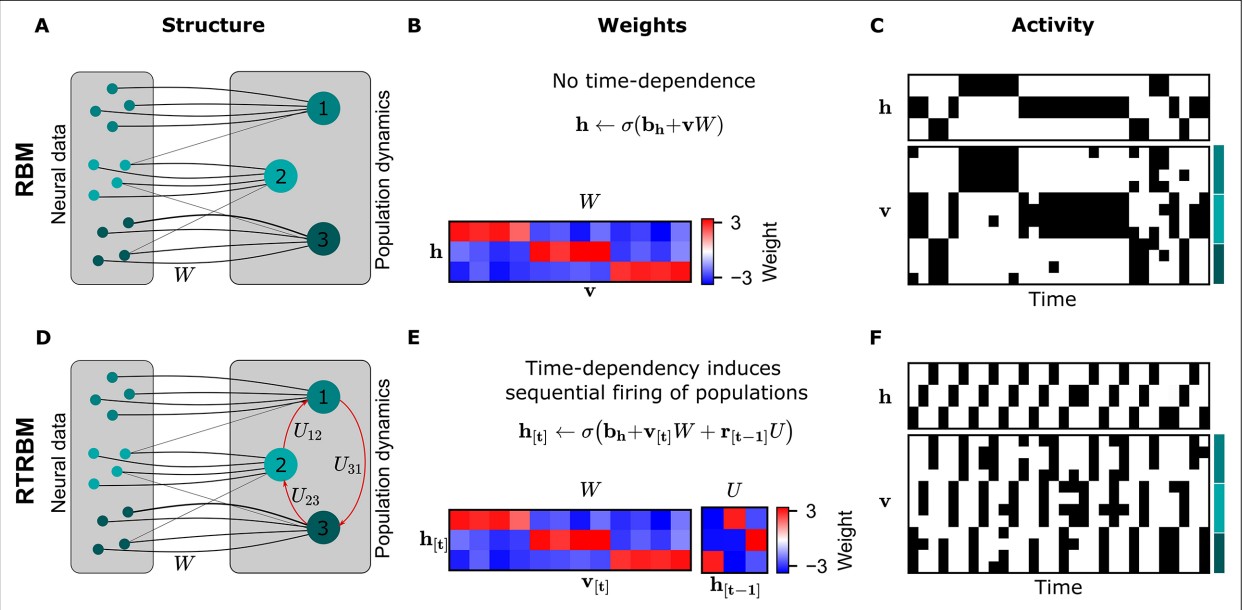

**Figure 1.** The recurrent temporal RBM (RTRBM) extends the restricted Boltzmann machine (RBM) by additionally accounting for temporal interactions of the neural assemblies. (**A**) A schematic depiction of an RBM with visible units (neurons) on the left, and hidden units (neural assemblies) on the right. The visible and hidden units are connected through a set of weights $W$. (**B**) An example $W$ matrix where a subset of visible units is connected to one hidden unit. Details of the equations in panel B and E are given in Materials and methods. (**C**) Hidden and visible activity traces were generated by sampling from the RBM. Due to its static nature, the RBM samples do not exhibit any sequential activation pattern, but merely show a stochastic exploration of the population activity patterns. (**D**) Schematic depiction of an RTRBM. The RTRBM formulation matches the static connectivity of the RBM, but extends it with the weight matrix $U$ to model temporal dependencies between the hidden units. (**E**) In the present example, assembly 1 excites assembly 2, assembly 2 excites assembly 3, and assembly 3 excites assembly 1, while the remaining connections were set to 0. (**F**) Hidden and visible activity traces were generated by sampling from the RTRBM. In contrast to the RBM samples, the RTRBM generates samples featuring a sequential firing pattern. It is able to do so due to the temporal weight matrix $U$ which enables modeling temporal dependencies.

## The RTRBM extends the RBM by temporal dynamics on the assembly level

The principal structural difference between the RBM and the RTRBM is the addition of recurrent connections through a set of weights $U$ in the RTRBM that connect the hidden unit states at time-steps $t-1$ and $t$. These connections allow the RTRBM to incorporate temporal dynamics from the data, while the (c)RBM is only able to represent time-independent, statistical relationships. This difference is illustrated in a small example of neural assemblies in *Figure 1*.

When applying the RBM to neural data, the neurons are represented by the *visible units*, while the underlying neural assemblies are represented by the *hidden units* (see *Figure 1* for a visualization). In this basic example, we connect each assembly to an exclusive set of neurons for simplicity (*Figure 1A*). In the case of the RTRBM, there are additional, direct connections between the assemblies (*Figure 1D*, red arrows, defined by the weights $U$). To emphasize the resulting difference in temporal dynamics, we initialize an RBM and an RTRBM model with matching hidden to visible connections $W$ (*Figure 1B and E*). We set these connections up so that each neural assembly is connected to a single hidden unit. In this manner, the hidden unit dynamics act as assembly dynamics, where each hidden unit represents a distinct assembly. In addition, the RTRBM has a weight matrix $U$ that models the inter-assembly temporal dynamics. The activation function applied to the sum of the inputs is a sigmoid function, which enforces the output to be in the range $[0, 1]$. The weights that connect the visible to the hidden units outside their assigned assembly must, therefore, be negative to prevent their participation in other assemblies.

To demonstrate the resulting difference in temporal dynamics, we sample from the RBM and RTRBM and compare the resulting activity traces. In the RBM, as expected from the model definition, the stochastic sampling between the hidden and visible units does not lead to systematic sequential

activations of the assemblies occur, aside from some persistence due to the reactivation of similar ensembles through the weights $W$.

Conversely for the RTRBM, a combination of temporal sequences and stochastic exploration can be realized: In this simple example of an RTRBM, assembly 1 excites assembly 2, assembly 2 excites assembly 3 and assembly 3 again excites assembly 1 (see *Figure 1E*). As a result, the hidden activity traces of data sampled from the RTRBM show matching sequential activation of hidden units (*Figure 1F*). Because each hidden unit is connected to a subset of visible units, the results in a sequential activation of the assemblies, where typically only one assembly is strongly active at each time-step. As the representation of the RTRBM is still probabilistic, the dynamics display a mixture of dynamic, and stochastic properties, which we consider a hallmark of neural activity.

In summary, the RTRBM maintains the features of the RBM to provide an interpretable and probabilistic representation of neural data, but extends it to include temporal dependencies between neural assemblies.

## RTRBMs learn assembly dynamics from simulated neural data

In the above example, the neural assembly connectivity was predefined. Next, we demonstrate that the RTRBM can be trained on simulated neural data to learn a set of weights $W$ and $U$ that correctly captures the underlying temporal dynamics on the assembly level. Initially, we aimed to compare the performance of the cRBM with the cRTRBM. However, we did not manage to get the RTRBM to reach the compositional phase. To ensure a fair and robust comparison, we opted to compare the RBM with the RTRBM. In this test case, we indeed find the RTRBM to outperform the RBM in the representation of the underlying moments.

We devised a method for generating artificial data sets mimicking neural population activity using a simplified neural network model. Here, neural activity is driven by the population activity of underlying neural assemblies. These activities of assemblies were determined by two factors: endogenous, assembly-specific activations, and recurrent activations through the connections between assemblies (*Figure 2A*, left). The activity of the neurons was then generated from a Poisson process whose time-dependent rate was given by the activations of a single assembly population. For clarity of the presentation, we here again implement a direct match between assemblies and neurons, and thus expect the estimated weight matrix to be a 'diagonal' matrix between assemblies and neurons.

An RBM (*Figure 2B*) trained on the simulated data recovers a non-diagonal weight matrix (*Figure 2C*), which is composed of both the true assembly-to-neuron weights on the diagonal, but in addition has multiple off-diagonal weights, which partially account for the dependencies between the assemblies.

In contrast, the RTRBM correctly segments all ten assemblies, recovering a clean 'diagonal' estimated connectivity matrix $\hat{W}$ (*Figure 2C*, right), in addition to providing a close estimate ($\hat{U}$) to the true assembly connectivity matrix $U$, i.e., it recovers the underlying hidden connections from the activation patterns of neurons alone. Consistently, each visible neuron has only a single dominant weight ($|\hat{W}_{ij}|$) in the RTRBM and thus produces a diagonal weight matrix, while the RBM assigns multiple strong weights to an RBM to address the time-dependencies (*Figure 2D*).

As expected, the RBM performs very well in capturing the average activations of the visible units ($\langle v_i \rangle$) and their correlations ($\langle v_i v_j \rangle$), referred to as first and second order moments, respectively (*Figure 2E*, top). However, it cannot accurately capture the time-shifted moments of the visible or hidden units (*Figure 2E*, bottom). The RTRBM performs similarly for the simultaneous moments (*Figure 2E*, top), but provides a more accurate account of the time-shifted moments (*Figure 2E*, bottom). This behavior is consistent for the different moments across multiple runs on independent simulated data sets and model estimates ($N = 10$, *Figure 2F*), with significant improvements of the RTRBM observed for the time-shifted moments (p-values 0.993 for $v_i$, 0.312 for $h_i$, $9.13 \cdot 10^{-5}$ for $v_i^t v_j^{t+1}$ and $4.55 \cdot 10^{-3}$ for $h_i^t h_j^{t+1}$, one-sided Mann-Whitney U test).

Lastly, the RTRBM also exhibited a significantly lower normalized mean squared error (nMSE, see Materials and methods) ($p = 4.1 \cdot 10^{-8}$, two-way ANOVA with time-steps and model type as factors, $N = 10$, *Figure 2G*) when predicting ahead in time inside the simulated data (not used in training). The RTRBM's advantage in prediction stayed significant up to four time-steps ($p < 0.001$, two group t-tests per time-step with Bonferroni correction for the number of time-steps). This decay of differences between the models is expected, as the probabilistic basis of the RBM/RTRBM as well as the

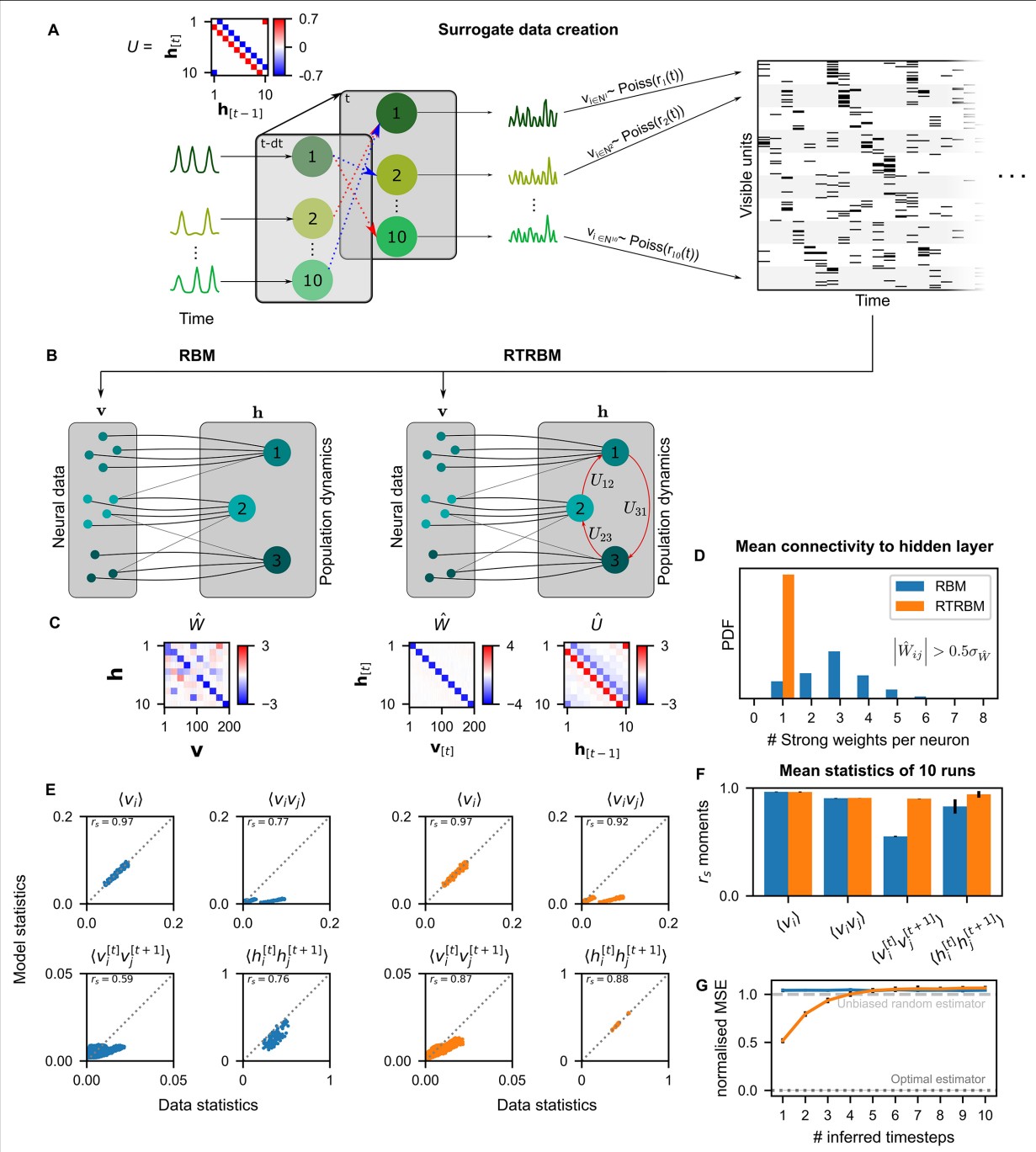

**Figure 2.** The recurrent temporal RBM (RTRBM) outperforms the restricted Boltzmann machine (RBM) on sequential statistics on simulated data. (**A**) Simulated data generation: Hidden Units ($N_h$) interact over time to generate firing rate traces which are used to sample a Poisson train. For example, assembly 1 drives assembly 2 and inhibits assembly 10, both at a single time-step delay. (**B**) Schematic depiction of the RBM and RTRBM trained on the simulated data. (**C**) For the RBM, the aligned estimated weight matrix $\hat{W}$ contains spurious off-diagonal weights, while the RTRBM identifies the correct diagonal structure (top). For the assembly weights $U$ (left), the RTRBM also converges to similar aligned estimated temporal weights $\hat{U}$ (right). (**D**) The RTRBM attributes only a single strong weight to each visible unit (($w_{i,j} > 0.5\sigma$, where $\sigma$ is the standard deviation of $W$)), consistent with the specification in $W$, while in the RBM multiple significant weights get assigned per visible units. (**E**) The RBM and RTRBM perform similarly for concurrent ($\langle v_i \rangle$, $\langle v_i v_j \rangle$) statistics, but the RTRBM provides more accurate estimates for sequential ($\langle v_i^{[t]} v_j^{[t+1]} \rangle$, $\langle h_i^{[t]} h_j^{[t+1]} \rangle$) statistics. In all panels, the abscissa refers to the data statistics in the test set, while the ordinate shows data sampled from the two models,, respectively. (**F**) The trained RTRBM and the RBM yield similar concurrent moments, but the RTRBM significantly outperformed the RBM on time-shifted moments (see text for details on statistics). (**G**) The RTRBM achieved significantly lower normalized mean squared error (nMSE) when predicting ahead in time from the current state in comparison to RBMs for up to four time-steps.

*Figure 2 continued on next page*

*Figure 2 continued*

The online version of this article includes the following figure supplement(s) for figure 2:

**Figure supplement 1.** Alignment of weight matrices after learning.

simulated data by design leads to non-deterministic trajectories, similar to the divergence of trajectories in non-linear dynamic systems where small noise eventually leads to large differences (*Strogatz, 2000*) (see discussion for a relation to animal behavior).

These results indicate that the RTRBM provides a more accurate account of the model structure and data statistics in particular for sequential activations. The RBM can partially account for the temporal structure, but only by conflating it with its non-temporal weights in $W$.

## The RTRBM outperforms the cRBM on whole-brain zebrafish data

Next, we trained the RTRBM on whole-brain data recorded in zebrafish larvae ($n = 8$, same data as in *van der Plas et al., 2023*). To obtain binarized spike traces that can be used by the RTRBM, the individual fluorescence traces were deconvolved by means of blind sparse deconvolution (*Figure 3A*). Model training was performed using a trained cRBM as the basis for the assembly-to-neuron weights $W$, and then training the temporal assembly-to-assembly weights $U$, while allowing $W$ to only change slightly (the learning rate for these weights is reduced by two orders of magnitude, see Materials and methods for more details on model training). For each animal, model training was successful and the weight changes converged to small values. This approach of using pre-learned weights can be seen as a variant of transfer learning (*Tan et al., 2018*). We chose for this training procedure as the weight matrix $W$ inferred by the RTRBM is rarely able to identify localized receptive fields for a large portion of hidden units within its present, non-compositional, formulation.

The mean square reconstruction error $\frac{1}{N} \sum_{i=1}^{N} (\mathbf{v}_{\text{data},i} - \mathbf{v}_{\text{model},i})^2$ in the initial phase of model training was $\sim 0.40$, the RTRBM was able to reduce this to $\sim 0.072$, which it achieves predominantly by adjusting the temporal weights. The trained RTRBM model maintained the localised neural assemblies inherited from the cRBM (*Figure 3C*) as quantified by a sparse weight distribution (*Figure 3B*, left) and a comparable, lower number of typically 1–3 strong weights per neuron as in the cRBM (*Figure 3B*, right).

Most hidden units showed self-excitation, i.e., indicated as a positive value on the diagonal of $\hat{U}$. The overall pattern of temporal connections between the assemblies in $\hat{U}$ could be divided into several groups. To this end, agglomerative clustering (for details see Materials and methods) can be applied to the incoming (row) or outgoing (columns) connections of the matrix $\hat{U}$ to identify assemblies with similar temporal structures. We here focus on the incoming connections/receptive fields, as their grouping was more clear (*Figure 3D*, dashed lines indicate boundaries between clusters). The clustering for outgoing connections was similar, however, not identical as $\hat{U}$ are generally not symmetric, due to the directedness of the temporal connections. The identified clusters showed characteristic patterns of connectivity, e.g., clusters 1 and 5 show a diverse connectivity pattern with relatively strong intra-cluster connections (*Figure 3D*). Clusters 2 and 3 exhibit a diverse connectivity pattern as well, but do not show the same strong intra-cluster connectivity. Cluster 4 has very strong recurrent intra-cluster connectivity, but also excites all other clusters. Cluster 5 is dominated by strong inhibitory connectivity to itself and all other clusters. We thus see a range of different connectivity patterns appearing, identified by the clustering method. By using a lower clustering threshold, an even further refined clustering structure appears (data not shown here). The sets of strongly connected visible units corresponding to the hidden unit clusters furthermore formed spatially localized sets of neurons (*Figure 3E*).

To compare the performance between the inferred RTRBM and cRBM models, we analyzed the reconstruction quality. To this end, we sampled data from both models and again compared the model statistics to the data statistics, using an unseen test set (matched to the test set in *van der Plas et al., 2023*, see Materials and methods for details). For the example fish in *Figure 3B–F*, the first order moments between neurons $\langle v_i \rangle$ are strongly correlated for the RTRBM ($r_s = 0.92$, $p < \epsilon$, where $< \epsilon$ denotes machine precision), that is an improvement compared to the performance of the cRBM ($r_s = 0.75$, $p < \epsilon$). The second-order moments between neurons $\langle v_i v_j \rangle$ of the RTRBM ($r_s = 0.58$, $p < \epsilon$) also correlates better compared to the cRBM ($r_s = 0.27$, $p < \epsilon$). To establish how well both models can capture the temporal dynamics of the data, we compared the time-shifted moments of the visible

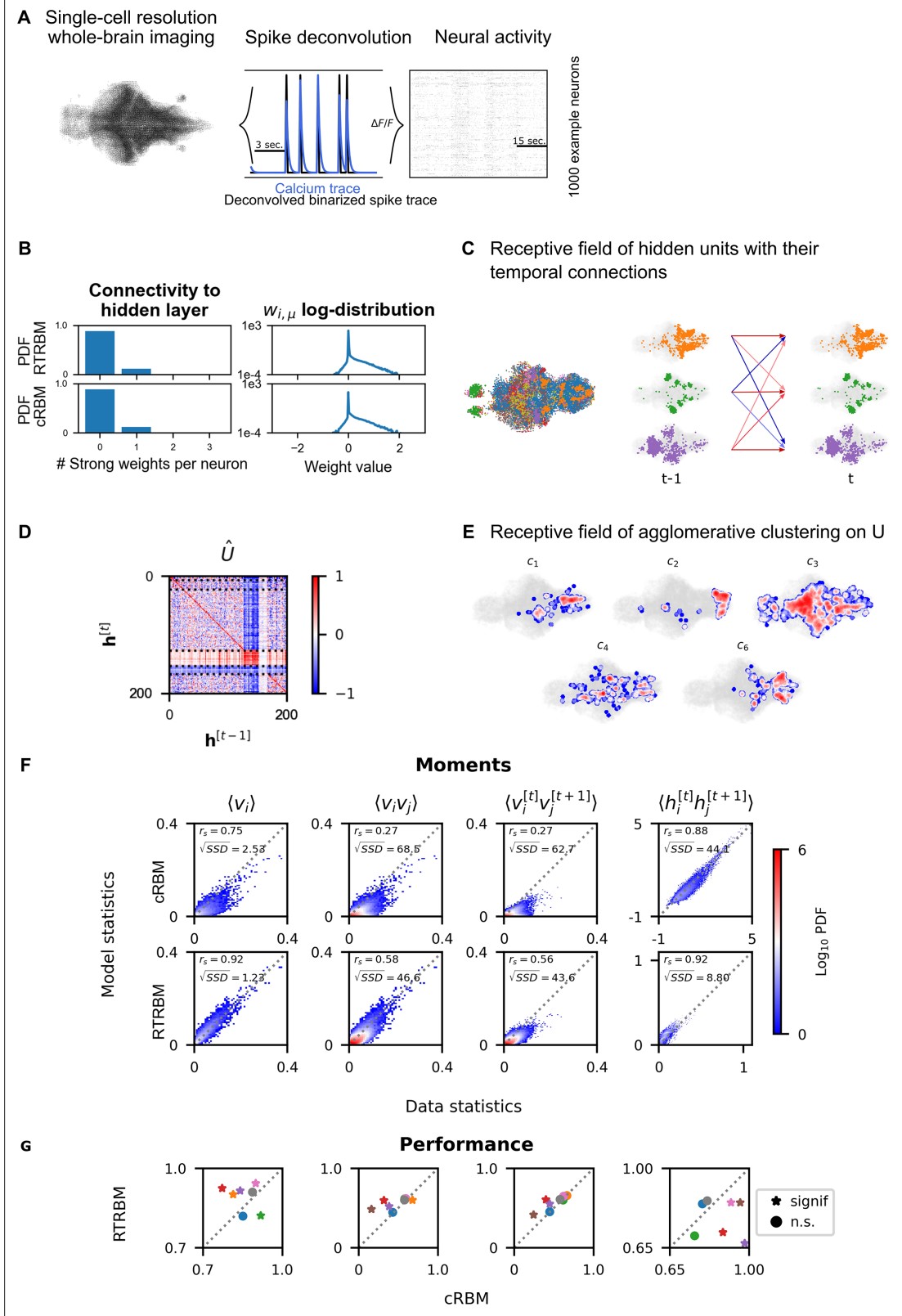

**Figure 3.** Recurrent temporal RBM (RTRBM) often outperforms the compositional restricted Boltzmann machine (cRBM) on zebrafish data. (**A**) Whole-brain neural activity of larval zebrafish was imaged via Calcium-indicators using light-sheet microscopy at single neuron resolution (left). Calcium activity (middle, blue) is deconvolved by blind, sparse deconvolution to obtain a binarized spike train (middle, black). The binarized neural activity of 1000 randomly chosen neurons (right). (**B**) Left: Distribution of all visible-to-hidden weights. Here, a strong weight is determined by proportional thresholding,

*Figure 3 continued on next page*

*Figure 3 continued*

$w_{i,j} > w_{thr}$. Here $w_{thr}$ is set such that 5000 neurons have a strong connection towards the hidden layer. Right: log-weight distribution of the visible to hidden connectivity. (**C**) The RTRBM extracts sample assemblies (color indicates assembly) by selecting neurons based on the previously mentioned threshold. Visible units with stronger connections than this threshold for a given hidden unit are included. Temporal connections (inhibitory: blue, excitatory: red) between assemblies are depicted across time-steps. (**D**) Temporal connections between the assemblies are sorted by agglomerative clustering (dashed lines separate clusters, colormap is clamped to $[-1, 1]$). Details on the clustering method can be found in Materials and methods. (**E**) Corresponding receptive fields of the clusters identified in (**D**), where the visible units with strong weights are selected similarly to (**B**). The receptive field of cluster 5 has been left out as it contains only a very small number of neurons with strong weights based on the proportional threshold. (**F**) Comparative analysis between the cRBM (bottom row) and RTRBM (top row) on inferred model statistics and data statistics (test dataset). Compared in terms of Spearman correlations and sum square difference. From left to right: the RTRBM significantly outperformed the cRBM on the mean activations $\langle v_i \rangle$ ($p < \epsilon$), pairwise neuron-neuron interactions $\langle v_i v_j \rangle$ ($p < \epsilon$), time-shifted pairwise neuron-neuron interactions $\langle v_i^{[t]} v_j^{[t+1]} \rangle$ ($p < \epsilon$), and time-shifted pairwise hidden-hidden interactions $\langle h_i^{[t]} h_j^{[t+1]} \rangle$ ($p < \epsilon$) for example fish 4. (**G**) The methodology in panel F is extended to analyze datasets from eight individual fish, each color representing one individual fish. Spearman correlation and the assessment of significant differences between both models are determined using a bootstrap method (see Materials and methods for details).

$\langle v_i^{[t]} v_j^{[t+1]} \rangle$ and hidden $\langle h_i^{[t]} h_j^{[t+1]} \rangle$ units. The time-shifted moments of the visible units of the RTRBM ($r_s = 0.56$, $p < \epsilon$) correlates better than the cRBM ($r_s = 0.27$, $p < \epsilon$). While a direct comparison of the hidden unit activations between the cRBM and the RTRBM is hindered by the inherent discrepancy in their activation functions (unbounded and bounded, respectively), the analysis of time-shifted moments reveals a stronger correlation for the RTRBM hidden units ($r_s = 0.92$, $p < \epsilon$) compared to the cRBM ($r_s = 0.88$, $p < \epsilon$).

The Spearman correlation is scale-free, i.e., even if one variable is doubled, the correlation can stay the same. However, in many cases, the sampled RTRBM was much closer to the test data in absolute terms (indicated by densities in *Figure 3F* that are closer to the diagonal). To quantify this difference, we also compared the sum square difference (SSD) between the sampled statistics of the RTRBM and cRBM with the statistics of the test set, to determine how well both models accounted for the real data on a quantitative, absolute level. The RTRBM had a lower SSD for all first-, second-, and time-shifted moments compared to the cRBM. This suggests that the statistics of the RTRBM are better matched in an absolute sense, and can be considered as better behaved than the statistics of the cRBM when compared to the test set.

Over the whole dataset, the RTRBM outperforms the cRBM on 5 out of 8 fish for $\langle v_i \rangle$, and on 4 out of 8 fish for $\langle v_i v_j \rangle$, while the performance was similar on the remaining fish, except for the green fish (*Figure 3G*). To establish how well both models can capture the temporal dynamics of the data, we compared the $\langle v_i^{[t]} v_j^{[t+1]} \rangle$ and $\langle h_i^{[t]} h_j^{[t+1]} \rangle$. Here, the RTRBM outperformed the cRBM on 4 out of 8 fish for the visible units and on 6 out of 8 fish for the hidden units. Specifically, the performance of the RTRBM is consistently improved compared to the cRBM for the same fish across the different moments (*Figure 3G*). Note, that the second order statistics are statistics the algorithms are not explicitly trained on to replicate (see also Materials and methods).

In summary, the transfer learning approach was able to successfully expand the cRBM model to include the temporal connections, while maintaining a high level of sparsity in the hidden-to-visible layer connections. It is beyond the scope of this study to evaluate the detailed differences between the two models, but this transfer learning approach appears a promising avenue to enable the RTRBM to be estimated on large scale data sets.

## Identification of the underlying time-scale of assembly interactions

The sampling rate in an experiment will generally not match the effective interaction time between neural assemblies. If the mismatch is too large, it may prevent the RTRBM from making a correct estimate of the temporal connections. It is therefore important to be able to estimate the interaction time of assemblies in relation to the sampling rate.

To investigate this issue, simulated data was generated where the interaction time between assemblies was set to $\Delta t_A = 4$ time-steps (relative to the sampling rate of the simulated data, *Figure 4A*, left). This simulated data was then down-sampled to different rates using integer steps on the range $[1, 10]$ (*Figure 4A*, middle). Down-sampling was performed by selecting the value of the data at the sampling interval, rather than averaging or interpolating over all points in the interval. This choice was motivated by the fact that light-sheet imaging only has access to the neural activity at particular

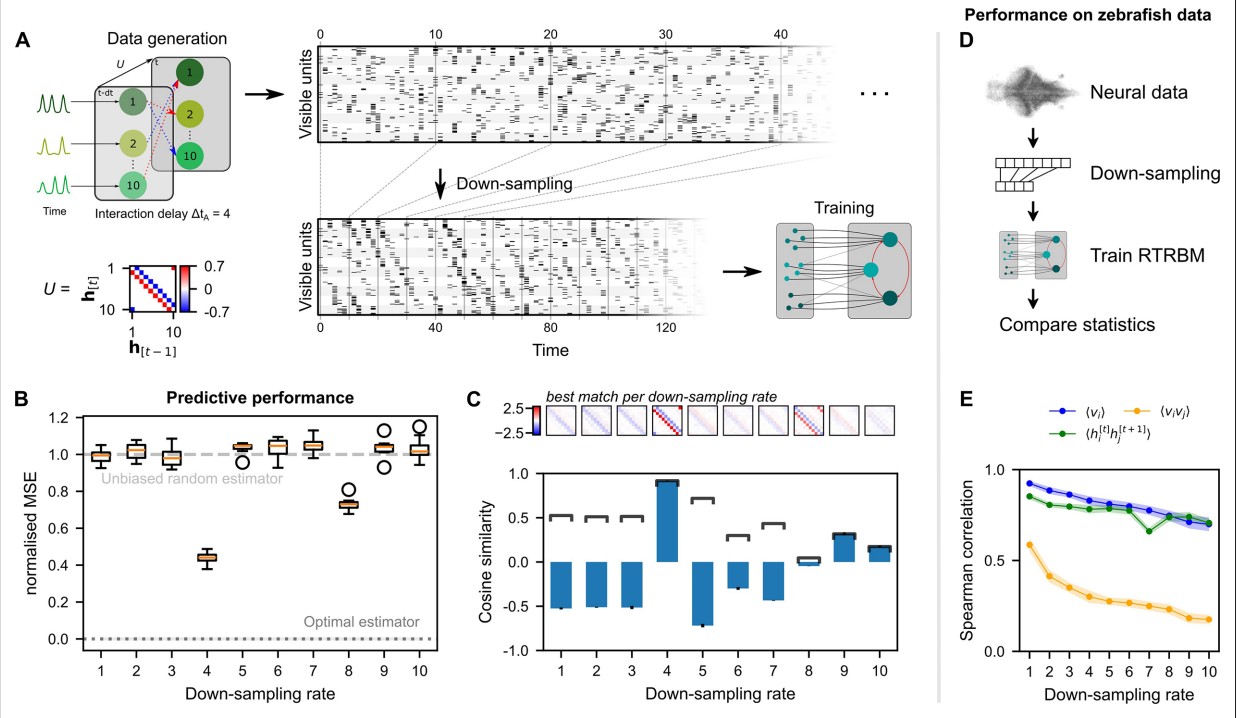

**Figure 4.** Neural interaction timescale can be identified via recurrent temporal RBM (RTRBM) estimates over multiple timescales. (**A**) Training paradigm. Simulated data is generated as in **Figure 2**, but with temporal interactions between populations at a delay of $\Delta t_A = 4$ time-steps. This data is downsampled according to a downsampling rate $\Delta t_D$ by taking every $\Delta t_D$-th time-step (shown here is $\Delta t_D = 4$), and used for training different RTRBMs. (**B**) Performance of the RTRBM for various down-sampling rates measured as the normalized mean squared error (MSE) in predicting the visible units one time-step ahead ($N = 10$ models per $\Delta t_D$). Dotted line shows the mean estimate of the lower bound ± SEM ($N = 10,000$) due to inherent variance in the way the data is generated (see Materials and methods). Dashed gray line indicates the theoretical performance of an uninformed, unbiased estimator $P\left(\hat{v}_i^t = 1\right) = \langle v_i \rangle$. (**C**) Cosine similarity between the interaction matrix $U$ and the aligned learned matrices $\hat{U}$, both z-scored. Bars and errorbars show mean and standard deviation, respectively, across the $N = 10$ models per $\Delta t_D$. Dark lines show absolute values of the mean cosine similarity. Shown above are the $\hat{U}$ matrices with the largest absolute cosine similarity per down-sampling rate. (**D**) The same procedure as in (**A**) is performed on neural data in order to find the effect of down-sampling here. (**E**) Spearman correlation of three important model statistics across different down-sampling rates for neural data from example fish 4, similar to **Figure 3F**. Dots and shaded areas indicate mean and two times standard deviation, determined using a bootstrap method (see Materials and methods for details).

time-points and cannot average the entire duration between these time-points (as it is imaging at different planes in depth in between). As the size of the system should not make a qualitative difference for this analysis, we generated simulated data with only $N_h = 10$ hidden assemblies and $N_v = 20$ visible units per assembly. Different runs ($N = 10$) were inferred on independently drawn assembly dynamics and subsequently drawn spike-times, but with identical temporal and static weight matrices $W$ and $U$.

To evaluate the reconstruction performance of the trained RTRBMs, the reconstruction nMSE one time-step ahead is calculated on a single test data set and compared for different down-sampling rates (**Figure 4B**, see Materials and methods for details). The trained RTRBM performs significantly better than an unbiased random estimator, i.e., $P\left(\hat{v}_i^t = 1\right) = \langle v_i \rangle$ (nMSE = 1), when the down-sampling rate is a multiple of the simulated interaction time ($p = 0.0098$ for $\Delta t_D \in \{4, 8\}$, one-sided Wilcoxon signed rank test, $N = 10$ with Bonferroni correction for the number of down-sampling rates, effect size $\geq 8.93$). The performance is best when the down-sampling factor matches this interaction time at $\Delta t_D = 4$ ($p = 0.00082$, one-sided Wilcoxon rank-sum test, $N = 10$ with Bonferroni correction for the number of comparisons, effect size $\geq 7.05$). The optimal estimator at nMSE = 0 is obtained from knowing the underlying model precisely, only limited by the unpredictable variance from the random factors of Poisson sampling and intrinsic assembly dynamics (see Materials and methods for details). Visual inspection of the inferred data shows that only the sampled data from $\Delta t_D = 4$ contains the characteristic temporal sequences generated by the connections in $U$.

To verify that the correct temporal connections between the hidden units are identified, the estimated temporal weights $\hat{U}$ after alignment of assemblies (see Materials and methods) are compared with the true temporal connections $U$ of the simulated data using cosine similarity (*Figure 4C*). Correspondingly, the similarity peaks when the down-sampling rate matches the interaction time. While neighboring values around the simulation step size have similar absolute correlations, the correct step size still outperforms them. Conversely, this indicates that it may be sufficient to be close to the true step size in order to correctly estimate the temporal dependence in $U$.

We applied this analysis as a refinement to the training of the RTRBM on zebrafish data, and found that the Spearman correlation of important moments between neural activity and model activity was highest at the natural sampling rate (*Figure 4E*). Therefore, the analysis in *Figure 3* was conducted without down-sampling. Each RTRBM was trained with the same number of gradient updates to ensure a fair comparison. However, due to the down-sampling procedure the amount of training data available is drastically decreased for large down-sampling rates. We did not retrain the cRBM on the downsampled data because the cRBM model does not account for time dependencies.

## Discussion

Here, we introduced the RTRBM as a powerful dynamical statistical model for the analysis of large-scale neural data, demonstrating that it can uncover temporal dependencies between neural assemblies. We achieve this through transfer learning on the basis of the static assembly structure estimated by a cRBM trained on the same data (*van der Plas et al., 2023*). The estimated RTRBM models are structurally more fitting and provide more accurate accounts of the activity dynamics than those of the cRBM, as we demonstrate on simulated and experimentally acquired, whole-brain zebrafish data. The resulting temporal connectivity structure on the assembly level provides a compact description of the neural dynamics, which decomposes into dynamical networks of assemblies. Training of an RTRBM/cRBM model can be completed in a few hours on current hardware, and could thus lend itself for within experiment, interventional studies. The RTRBM therefore provides an effective and practically feasible model formalism for accounting for temporal dynamics as well as stochastic properties of whole-brain zebrafish activity.

### Relation with previous studies on large-scale assemblies

Estimating functional divisions and connectivity from large-scale activity data can be considered one of the key objectives of computational neuroscience, as it would allow to automatically extract interpretable structure from datasets of (human-level) uninterpretable complexity. While it is generally recognized that this poses a difficult analytical challenge, in particular in highly connected systems (*Das and Fiete, 2020*), whole-brain recordings have brought a critical advance to this endeavor. However, due to the relatively recent introduction of whole-brain measurements in zebrafish larvae (*Ahrens et al., 2013*), surprisingly few studies exist in this system that have attempted the investigation of neural activity in this type of data at the whole-brain scale (*Nguyen et al., 2018*; *Chen et al., 2018*; *Betzel, 2020*; *van der Plas et al., 2023*).

These studies have all focused on extracting functional groupings from the neural activity, without directly attempting to perform temporal predictions of neural activity. In *Chen et al., 2018*, a clustering approach was introduced that identified a set of clusters of neurons, which showed responses to specific visual stimuli or motor behaviors. (*Betzel, 2020*) estimates instantaneous functional connectivity from spontaneous activity and identifies groups of local nodes that form a hierarchical, modular structure, however, without the possibility of using this model in a generative way. In our previous study (*van der Plas et al., 2023*) we identified neural assemblies from spontaneous activity using a generative, probabilistic approach, i.e., the cRBM, but without explicitly modeling any time-dependencies. Lastly, (*Nguyen et al., 2018*) uses Gaussian mixture modelling to cluster on the activity level, but again, this method does not yield any insight into the time-dependencies between the identified clusters.

Many other studies have focused on the analysis of subsystems, but also without directly modelling the temporal dependence, e.g., sensorimotor transformations in the visual (*Bianco and Engert, 2015*) and the auditory system (*Privat et al., 2019*), the representation and maintenance of spatial location

(*Yang et al., 2022*), decision making (*Bahl and Engert, 2020*), or the neural circuit underlying heading direction (*Petrucco et al., 2023*).

Since the current temporal resolution of light-sheet imaging is rather low, i.e., limited of a few volumes per second (but alternative scanning approaches might change this soon, see e.g., *Bouchard et al., 2015*), estimating the temporal dynamics on the level of individual neurons is still difficult. Therefore, our present approach focuses on the dynamics between assemblies, which are expected to develop on slower timescales. Temporal connections are generally more insightful than instantaneous function connections, as they are directed and therefore provide a better basis for separating correlation from causation.

We demonstrate that the RTRBM was able to capture functional temporal connections between neural assemblies while maintaining localized receptive fields of the hidden units. Additionally, as the RTRBM yields insight into the temporal dynamics of these identified neural assemblies, it provides a way of identifying which assemblies are similar in their dynamics and thereby can suggest distinct large-scale dynamical networks spanning one or multiple brain areas (see *Figure 3*). Moreover, the RTRBM outperforms the classical RBM on the artificial data in terms of reconstruction statistics, and also outperforms cRBM in accounting for temporal dynamics.

To our knowledge, only two earlier studies have attempted to predict dynamics or estimate dynamical relations between neural assemblies on the whole brain level (*Watanakeesuntorn et al., 2020*; *Pao et al., 2021*). In both studies Empirical Dynamical Modelling/Convergent Cross Mapping is used to estimate neural dynamics, however, the zebrafish data is mostly utilized as a usage case for demonstrating that the methods scale to large datasets, without providing insight into the resulting ensembles or prediction quality. Another approach used is to apply dynamical modeling of the behavioral level and then use concurrently acquired whole-brain activity to identify corresponding structures in the zebrafish brain (*Dunn et al., 2016*).

## Limitations and future improvements

The RTRBM introduces temporal dependencies in a constrained way that effectively limit the number of additional parameters. This feature is important to avoid overfitting on the limited amount of data generally available in each experiment. However, the increased complexity involved with the addition of these temporal dependencies limits the analytical tractability of the model. This added complexity had a number of consequences on the estimation procedure, which should ideally be resolved in future work.

Specifically, the RTRBM in its current form is not intrinsically driven toward the compositional phase, which is an important property that pushes the model to identify localized neural assemblies. Specifically, the use of dReLU hidden unit potentials within the RTRBM framework was not analytically tractable in our hands. We therefore opted for a transfer learning approach, where the cRBM first estimates the assemblies, and then we initialize the model with these identified assemblies. The RTRBM then infers the temporal connectivity between the identified localized assemblies, while only slightly modifying their assembly structure. This approach could be limiting in multiple ways. For example, the cRBM-estimated assembly structure could contain an amalgam of static and dynamic connectivities (see *Figure 2* for simulated data). Furthermore, it might be necessary to estimate the assembly structure jointly with the temporal connectivities between them for optimal decomposition. Extending the current work, we aim to refine the RTRBM by introducing other sparsity constraints on the hidden-hidden connections (similar to *Mittelman et al., 2014*), or by realizing the compositional properties in the RTRBM to allow single-step, direct estimation.

Another limitation of the current RTRBM framework is that all assemblies are interacting on a single time-scale. While we have demonstrated (*Figure 4*) that a single time-scale can be identified through the estimation of and subsequent selection from multiple RTRBMs on different timescales, the more general case of multiple interaction time-scales between different assemblies remains unaddressed. Partly, this issue is alleviated by the compounding effect of temporal interactions over multiple time-steps, which, therefore, suggests to err on the side of shorter time-steps in estimation. In preliminary explorations we noticed that the estimated RTRBM generates alternating dynamics between clusters of assemblies (see *Figure 3*) on time-scales that are much longer than the single interaction step. In subsequent work, the RTRBM could be generalized to include multiple time-scales of interaction for different assemblies.

Related to the time-scale issue, light-sheet imaging is currently limited to ~100 Hz, which means that the ~30–40 imaging planes are sampled at only 2–4 Hz, depending on the specific system. At these low imaging rates it is likely that some assembly dynamics are missed or appear simultaneous. Improvements in the speed of stepping between imaging planes will increase the sampling rate per cell. Together with brighter fluorescent indicators (*Zhang et al., 2023*), this will provide a more reliable basis for estimating models that incorporate temporal dependencies.

## Conclusions

The RTRBM formalism is the logical next step in the analysis of whole-brain recordings, as it accounts for the static and dynamic aspects using a probabilistic formalism, which captures both the stochastic and deterministic aspects that are hallmarks of neural activity. Followup studies need to attempt to extend the RTRBM into the compositional phase directly, thus speeding up learning and ensuring matched assemblies and temporal connectivities (*Bargmann and Marder, 2013*). Recordings at higher temporal resolutions and for longer durations will be instrumental in allowing convergence of the *cRTRBM* (*Helmstaedter, 2015*). Together with advancements in computing hardware this should allow for interventional studies based on the estimated dynamics to directly verify the estimated temporal connectivity through modulation techniques such as optogenetic control or laser ablation.

## Materials and methods
### The RBM

The RBM (*Salakhutdinov et al., 2007*) is an undirected graphical model that defines a probability distribution over a set of binary visible units carrying the data configurations $\mathbf{v} \in \{0, 1\}^{N_v}$, and real-valued latent representations are given by a set of hidden units $\mathbf{h} \in \mathbb{R}^{N_h}$. In contrast to the definition of the classical Boltzmann Machine, the RBM has no direct couplings between pairs of units within the same layer, making it a bipartite graph that allows for more efficient model training. The joint probability distribution of the model is defined as:

$$P(\mathbf{v}, \mathbf{h}) = \frac{1}{Z} \exp(-E(\mathbf{v}, \mathbf{h})) = \frac{1}{Z} \exp(\sum_i b_{v_i} v_i - \sum_j \mathcal{U}_j\left(h_j\right) + \sum_{i,j} W_{i,j} v_i h_j), \tag{1}$$

where $W_{ij}$ are visible-to-hidden unit weights, $E$ is the global energy of the RBM, $Z = \sum_\mathbf{v} \int d\mathbf{h} e^{-E(\mathbf{v},\mathbf{h})}$ the partition sum, and $\mathcal{U}_j\left(h_j\right)$ the hidden unit potential. The choice of the hidden unit potential shapes the energy landscape of the model and thus states the hidden units in the the model can take, making it an important model choice. For a more detailed discussion on the choice of the shape of the hidden unit potential, we refer to our previous paper (*van der Plas et al., 2023*). In this work, we use the Bernoulli hidden unit potential, which is defined as $\mathcal{U}_j\left(h_j\right) = -b_{h_j} h_j$, where $\mathbf{b}_h$ is the hidden bias and $\mathbf{h} \in \{0, 1\}^{N_h}$. Here, $i$ is used to index the visible units and $j$ to index the hidden units. The full set of model parameters is given by the visible bias $\mathbf{b}_v \in \mathbb{R}^{N_v}$, the hidden bias $\mathbf{b}_h \in \mathbb{R}^{N_h}$ and weights $W \in \mathbb{R}^{N_h \times N_v}$. Note that, while the visible units $\mathbf{v}$ are observed variables, the hidden units $\mathbf{h}$ are unobserved (latent) variables and must therefore be sampled, conditioned on the state of the visible units $\mathbf{v}$.

The posterior distributions over the hidden and visible units allow sequential sampling of the hidden and visible unit states. As both layers contain no within-layer dependencies, the posterior distributions over $\mathbf{v}$ and $\mathbf{h}$ are conditioned only on the other variable and factorize as:

$$P(\mathbf{h} \mid \mathbf{v}) = \prod_{j=1}^{M} P\left(h_j \mid \mathbf{v}\right) \propto \prod_{j=1}^{M} \exp\left(-\mathcal{U}_j\left(h_j\right) + h_j \cdot \sum_i W_{i,j} v_i\right),$$

$$P(\mathbf{v} \mid \mathbf{h}) = \prod_{i=1}^{N} P\left(v_i \mid \mathbf{h}\right) \propto \prod_{i=1}^{N} \exp\left(b_{v_i} v_i + v_i \cdot \sum_j W_{i,j} h_j\right) \tag{2}$$

The choice of the Bernoulli hidden unit potential reduces these equations to:

$$P\left(h_j = 1 \mid \mathbf{v}\right) = \sigma\left(b_{h_j} + \sum_i W_{ij} v_i\right)$$

$$P\left(v_i = 1 \mid \mathbf{h}\right) = \sigma\left(b_{v_i} + \sum_j W_{ij} h_j\right),$$

(3)

where $\sigma$ denotes the logistic function, defined as $\sigma(x) = \frac{1}{(1+\exp(-x))}$. For a detailed derivation, we refer to *Goodfellow et al., 2016*.

## Training the RBM

The RBM model parameters are learned by maximizing the log-likelihood of the target data, denoted as $\mathcal{L} = \langle \log(P(\mathbf{v})) \rangle_{\text{data}}$ (*Ackley et al., 1985*). This learning procedure ensures an accurate representation of the underlying distribution of the target dataset through the Boltzmann-Gibbs energy distribution (*Boltzmann, 1868*). Stochastic gradient-based methods are employed to minimize the Kullback–Leibler divergence (*Kullback and Leibler, 1951*) between the model and data distributions. These steps write:

$$\nabla_\theta \mathcal{L} = -\langle \nabla_\theta E(\mathbf{v}, \mathbf{h}) \rangle_{\text{data}} + \langle \nabla_\theta E(\mathbf{v}, \mathbf{h}) \rangle_{\text{model}},$$

(4)

where $E$ is the energy of the RBM, as given in probrbm. We here let $\langle ... \rangle$ denote the expectation value over the data and the model distributions, respectively. As the visible states of the data are given, $\langle E(\mathbf{v}, \mathbf{h}) \rangle_{\text{data}}$ is straightforward to compute. In contrast, the computation $\langle E(\mathbf{v}, \mathbf{h}) \rangle_{\text{model}}$ is intractable for sufficiently large systems due to the exponentially large state space over the visible unit states in the partition sum. To address this problem of intractability, a common solution is the use of a $K$-step Markov Chain Monte Carlo sampling scheme (Gibbs sampling) (*Hinton, 2002*). This approach starts from an initial configuration and aims to approximate the expectation values of the model distribution. This is also known as contrastive divergence (CD). In this learning scheme, samples are sequentially drawn from the visible and hidden posterior distributions, respectively (*Equation 4*), up to $K$ time. For sufficiently large values of $K$, this procedure yields unbiased samples of the underlying model distribution (*Hinton, 2002*). In practice, a value of $K = 1$ during training is generally sufficient to obtain reasonable expectation values (*Carreira-Perpinan and Hinton, 2005*). Increasing the number of samples will result in better approximations, and thus generally in improved model estimation.

## The compositional phase

In the classical definition of the RBM training scheme, there is no regularization on the hidden-unit activation sparsity. This means that a visible unit can have a proportionally strong connection to a large set of hidden units. Such a non-localized visible-hidden unit connectivity can hinder the interpretation of the model's learned latent representation. Previous work (*Tubiana and Monasson, 2017*; *Tubiana et al., 2019a*) introduced a method to regularize the RBM such that it is pushed towards a sparse visible-hidden unit connectivity, termed the compositional phase. The resulting cRBM was employed to discover compact neural assemblies in our previous study (*van der Plas et al., 2023*). This sparsity in connections allows for a direct and interpretable relationship between the weights in the cRBM and the generated data configurations. The compositional phase is observed when RBMs are constrained to a specified set of structural conditions (*Tubiana and Monasson, 2017*):

1. The hidden units are unbound and real-valued with ReLU like activation function.
2. The weight matrix $W$ is sparse.
3. The columns of the $W$ have similar norms.

A key implementation detail of this cRBM model is the use of the double-Rectified Linear Unit (dReLU) defining the hidden-unit potential (*Tubiana and Monasson, 2017*; *Tubiana et al., 2019a*; *Tubiana et al., 2019b*). Detailed analyses of RBMs operating in the compositional phase have exemplified the dynamical consequences and the advantages of this regime for learning complex data manifolds. In-depth discussions on the cRBM and the compositional phase can be found in related literature (*Tubiana and Monasson, 2017*; *Tubiana et al., 2019a*; *Tubiana et al., 2019b*). In this work, we use the implementation from *van der Plas et al., 2023* for our cRBM model. A detailed description of the implementation and accompanying code can be found there.

## The recurrent temporal restricted Boltzmann machine (RTRBM)

The RTRBM can be conceptualized as a series of RBMs unfolded across the temporal dimension, i.e., a recurrent network. The model state of the RTRBM at each time-step $t$ is essentially an RBM, where the hidden state is conditioned on the contextual hidden state of the RBM at time-step $t-1$. Mathematically, this involves augmenting the hidden bias of the RBM at time $t$ with an additional term dependent on the previous expected hidden states $\mathbf{r}^{[t-1]} = \left\langle \mathbf{h}^{[t-1]} \right\rangle$. Furthermore, the weights $U \in \mathbb{R}^{N_h \times N_h}$ are introduced which directly connect the previous to the current hidden unit states. This setup allow the model to directly capture latent-state temporal dependencies that cross multiple time-steps. The statistical distribution of the RTRBM at time-step $t$ is given by:

$$P\left(\mathbf{v}_{[t]}, \mathbf{h}_{[t]} \mid \mathbf{r}_{[t-1]}\right) = \exp\left(\mathbf{v}_{[t]}^\top b_v + \mathbf{h}_{[t]} W \mathbf{v}_{[t]}^\top + \mathbf{h}_{[t]}^\top \left(b_h + U\mathbf{r}_{[t-1]}\right)\right) / Z_{\mathbf{r}_{[t-1]}}, \tag{5}$$

where we dropped the indexing subscripts $ij$ for conciseness. At $t = 1$, the term $U\mathbf{r}_{[t-1]}$ is replaced by $\mathbf{b}_{\text{init}} \in \mathbb{R}^{N_h}$, which denotes the learnable initial bias for the hidden units. The joint probability distribution of an RTRBM model with length $T$ is found by factoring over the RBM stack over all time-steps $t, \ldots T$. This procedure is written as:

$$P\left(\left\{\mathbf{v}_{[t]}, \mathbf{h}_{[t]}\right\}_{t=1}^T \mid \left\{\mathbf{r}_{[t]}\right\}_{t=1}^{T-1}\right) = \frac{\exp\left\{E\left(\left\{\mathbf{v}_{[t]}, \mathbf{h}_{[t]}\right\}_{t=1}^T \mid \left\{\mathbf{r}_{[t]}\right\}_{t=1}^{T-1}\right)\right\}}{Z \cdot Z_{\mathbf{r}_1} \cdots Z_{\mathbf{r}_{[T-1]}}}, \tag{6}$$

where

$$
\begin{aligned}
E\left(\left\{\mathbf{v}_{[t]}, \mathbf{h}_{[t]}\right\}_{t=1}^T \mid \left\{\mathbf{r}_{[t]}\right\}_{t=1}^{T-1}\right) = & \\
-\left(\mathbf{h}_{[1]}^\top W \mathbf{v}_{[1]} + \mathbf{b}_v \mathbf{v}_{[1]} + \mathbf{b}_{\text{init}} \mathbf{h}_{[1]} + \sum_{t=2}^T \left(\mathbf{h}_{[t]}^\top W \mathbf{b}_{[t]} + \mathbf{b}_v \mathbf{v}_{[t]} + \mathbf{b}_h \mathbf{h}_{[t]} + \mathbf{h}_{[t]}^\top U \mathbf{r}_{[t-1]}\right)\right).
\end{aligned} \tag{7}
$$

Importantly, instead of using binary hidden unit states $\mathbf{h}_{[t-1]}$, sampled from the expected real-valued hidden states $\mathbf{r}_{[t-1]}$, the RTRBM propagates these real-valued hidden unit states directly. This approach constitutes the mean-field approximation of the hidden states of the temporally preceding RBMs, resulting in an efficient and easily computible approximation of the temporal state at each time-step $t$ (*Hinton, 2002*; *Sutskever et al., 2008*). The inputs $\mathbf{r}_{[t]}$ of the RTRBM at time-step $t$, given the visible unit state $v$, are then calculated as:

$$\mathbf{r}_{[t]} = \begin{cases} \sigma\left(\mathbf{W}_{[t]}\mathbf{v}_{[t]} + \mathbf{b}_h + U\mathbf{r}_{[t-1]}\right), & \text{if } t > 1 \\ \sigma\left(\mathbf{W}_{[t]}\mathbf{v}_{[t]} + \mathbf{b}_{\text{init}}\right), & \text{if } t = 1 \end{cases} \tag{8}$$

## Training the RTRBM

The model parameters $\theta \in \left\{W, U, \mathbf{b_v}, \mathbf{b}_{\text{init}}, \mathbf{b_h}\right\}$ are learned by maximization of the log-likelihood using (stochastic) gradient ascent:

$$\theta := \theta + \eta \nabla_\theta \mathcal{L}, \tag{9}$$

where $\eta$ denotes the learning rate and where

$$
\begin{aligned}
\nabla_\theta \mathcal{L} = & -\left\langle \nabla_\theta E\left(\left\{\mathbf{v}_{[t]}, \mathbf{h}_{[t]}\right\}_{t=1}^T \mid \left\{\mathbf{r}_{[t]}\right\}_{t=1}^{T-1}\right)\right\rangle_{\text{data}} \\
& + \left\langle \nabla_\theta E\left(\left\{\mathbf{v}_{[t]}, \mathbf{h}_{[t]}\right\}_{t=1}^T \mid \left\{\mathbf{r}_{[t]}\right\}_{t=1}^{T-1}\right)\right\rangle_{\text{model}}
\end{aligned} \tag{10}
$$

are the partial derivatives of the log-likelihood with respect to the model parameters. The energy function of the RTRBM can be split up into two components: a static component $\mathcal{H}$ and a temporal component $\mathcal{Q}$. The gradients of the static component $\mathcal{H}$ with respect to $\theta$ are calculated by summing over the gradients of the RBM at each time-step. The calculation of the gradients of $\mathcal{Q}$ with respect to $\theta$ is more complex. First, observe that $\mathcal{Q}$ can be computed recursively as:

$$\mathcal{Q}_{[t]} = \sum_{\tau=t}^{T} \mathbf{h}_\tau^\top U \mathbf{r}_{[\tau-1]} = \mathcal{Q}_{[t+1]} + \mathbf{h}_{[t]}^\top U \mathbf{r}_{[t-1]} \tag{11}$$

Hence, the backpropagation-through-time algorithm (**Rumelhart et al., 1986**) can be employed to recursively compute gradients for $\mathcal{Q}$ with respect to $\mathbf{r}_{[t]}$:

$$\nabla_{\mathbf{r}_{[t]}} \mathcal{Q}_{[t+1]} = U^\top \left( \nabla_{\mathbf{r}_{[t+1]}} \mathcal{Q}_{[t+2]} \odot \mathbf{r}_{[t+1]} \odot \left(\mathbf{1} - \mathbf{r}_{[t+1]}\right) + \mathbf{h}_{[t+1]} \right) \tag{12}$$

where $\odot$ denotes the element-wise product. The final step to obtain the derivative with respect to $\theta$ involves applying the chain rule:

$$\nabla_\theta \mathcal{Q} = \sum_{t=2}^{T} \left( \nabla_{\mathbf{r}_{[t]}} \mathcal{Q}_{[t+1]} \odot \nabla_\theta \mathbf{r}_{[t]} + \nabla_\theta \left( \mathbf{h}_t^\top U \mathbf{r}_{t-1} \right) \right). \tag{13}$$

For a more detailed derivation of these equations, we refer to **Mittelman et al., 2014**.

## Inference in the RTRBM

The RTRBM's sequential sampling scheme, using preceding hidden- and visible states to predict subsequent time-steps, enables generating data from its learned model distribution. Initially, $\mathbf{r}_{[t]}$ is calculated using the previous expected hidden states and current visible states. The contrastive divergence sampling scheme is then used to sample $\mathbf{v}_{[t+1]}$. Consequently, $\mathbf{v}_{[t+1]}$ is utilized to get the following expected hidden unit states $\mathbf{r}_{[t+1]}$, and so forth. This sequential process enables the RTRBM to statistically predict events in future time-steps.

During model training, all time-steps $T$ are available and it is not required to infer longer sequences. At the start of each training epoch $\mathbf{r}_{[t]}$ can be computed recursively up to time point $T$, followed by contrastive divergence performed for all time-steps in parallel. At model inference, we initialize $\mathbf{r}_{[t]}$ on the first time-step of each test batch and sample multiple time-steps into the future using the Gibbs sampling scheme (see **Figure 2G**). By comparing the inferred data with the remainder of the test data batch, we quantify the extent to which the RTRBM accurately captures the statistics of neural data across time (see also Performance metrics).

## Model-generated simulated data

The simulated data is generated according to the following principles: $N_h$ hidden units represent population activities that are temporally connected through a set of weights $U$. The activity of a single hidden unit is represented as a time-varying firing rate. Each hidden unit activity is the combination of two sources: (1) An intrinsic, time-varying firing rate, generated by two instances of randomly timed peaks (where the inter-peak interval is randomly drawn as ISI $\sim$ Uniform $(t_1, t_2)$) convolved with normalized Gaussian-shaped signals ($\phi(x; t, \sigma)$, where $\sigma \sim$ Uniform$(\sigma_1, \sigma_2)$ and where $\phi(x)$ signifies the standard Gaussian cumulative distribution function). Both of the resulting instances are subsequently renormalized to be in the intervals $[0, f_{max}/10]$ and $[0, f_{max}]$, respectively, and are summed together. The final signal is denoted $\lambda_i^{init}(t)$ and represents the firing rates of hidden unit $i$. (2) Recurrent interactions of firing rates between the hidden units at a delay of $\Delta t_A$, according to

$$\lambda_i(t) = \phi \left( \lambda_i^{init}(t) + \sum_{j=1}^{N_h} U_{ij} \lambda_j(t - \Delta t_A) \right) \tag{14}$$

where $\phi$ limits the output to the interval $[0, 3f_{max}]$. The strength of the temporal interactions between hidden units are given by the entries in $U$, with positive and negative entries representing excitatory and inhibitory interactions, respectively.

Each hidden unit connects to a distinct set of $N_{v\,per\,unit}$ visible units, yielding a total of $N_v = N_h \cdot N_{v\,per\,unit}$ visible units, and their activity $v_i$ is independently drawn from a Poisson distribution whose rate over time is given by the rate $\lambda_h$ of the corresponding hidden unit, scaled by a different constant for each visible unit taken from the range $(c_1, c_2)$. The first $t_{settle}$ time points are removed from the simulated data as the initial activity is different from the long-term interacting dynamics. The parameters for the data generating model were set to $N_h = 10$, $N_{v\,per\,unit} = 20$, $t_1 = 5$, $t_2 = 10$, $\sigma_1 = 0.1$, $\sigma_2 = 0.5$, $c_1 = 0.6$,

$c_2 = 1.4$, $t_{\text{settle}} = 25$, and $f_{\text{max}} = 0.8$. The temporal connectivity matrix $U$ was configured such that each population is both excited and inhibited by one of the other populations, and scaled in such a way that the resulting activity is mainly determined by the interactions (*Figure 2A* shows a graphical representation of the data generation pipeline).

For model training and evaluation, the generated simulated data was divided into a train- and test-set. Model inference is used for generating samples from trained RBM/RTRBM models to evaluate their performance. At model inference, the models are initialized on the first time-step of the test batch, and the subsequent time-steps are inferred through Gibbs sampling. The inferred data is then compared to the ground-truth test data through several statistics: (1) The MSE between inferred and test data, which measures how well the models can reproduce unseen activity of visible units. (2) The mean activation of the visible units $\langle v_i \rangle$, also referred to as the first order moments. (3) Pairwise moments between the visible units $\langle v_i v_j \rangle$, also denoted as second-order statistics. These pairwise moments assess the model's ability to capture data statistics it is not explicitly trained on. (4) Time-shifted pairwise moments of the visible and hidden units, i.e., $\langle v_i^{[t]} v_j^{[t+1]} \rangle$ and $\langle h_i^{[t]} h_j^{[t+1]} \rangle$, respectively. More details on how these statistics are calculated can be found in performance metrics.

## Zebrafish data

Whole-brain single-cell functional recordings for 15 zebrafish larvae were recorded by means of light-sheet microscopy in the lab of G. Debrégeas, 8 of which are used in this study (see *van der Plas et al., 2023* for a more detailed description of both the data acquisition process and subsequent data processing). These datasets are publicly available and can be found at https://gin.g-node.org/vdplasthijs/cRBM_zebrafish_spontaneous_data. In summary, the datasets consist on average of $40,709 \pm 13,854$ neurons, recorded for $5836 \pm 1183$ time-steps, at a frequency of 3.9 ± 0.8 Hz. The zebrafish larvae are 5–7 days post-fertilization and expressed GCaMP6s or GCaMP6f calcium reporters for imaging. The experimental procedure is described in more detail in *van der Plas et al., 2023*. The process from data segregation to analysis is displayed in *Figure 3*. To obtain binarized spike traces that can be used by the RTRBM the individual fluorescence traces are deconvolved by blind sparse deconvolution (*Figure 3A*; *van der Plas et al., 2023*).

## RTRBM initialization through transfer learning

Through empirical evaluation, we found de novo learning of the RTRBM to not converge to a solution that satisfies the compositional-phase criteria (see The compositional phase). This is to be expected as the formulation of the RTRBM as used in this work does not feature the required elements necessary to push the model solution towards such a solution. To overcome this issue, we make use of a transfer learning strategy (*Tan et al., 2018*). More specifically, the visible-to-hidden weights $W$ of the RTRBM models are initialized by their estimated counterparts from trained cRBM models (*van der Plas et al., 2023*). During subsequent training of the RTRBM, we let the model to update the values of these weights with a reduced learning rate (see below). This initialization strategy is able to bias the resulting weight matrix $W$ inferred by the RTRBM to contain a strong prior towards the localized receptive fields of the hidden units of the cRBM.

The hyperparameters, including the total number of hidden units for the cRBM model, were optimized by evaluating the model's performance over a grid of hyperparameter values using one dataset. This process identified the optimal hyperparameter values, which were subsequently applied to all recordings. In our study, the number of hidden units is fixed in the RTRBM model due to the use of transfer learning. Further details on the cross-validation process can be found in *van der Plas et al., 2023*.

For model training and evaluation, the functional data for each animal was divided into a train- and test-set. To this end, the functional recordings of each animal were subdivided into 10 segments of consecutive time-steps. In all cases, temporal segments 2, 6, and 7 are labeled as test sets, the remaining temporal segments are labeled as training sets. This data division strategy mirrors that of *van der Plas et al., 2023*. In practice, each segment was further subdivided into smaller batches of size $16 \pm 4$, enabling their computation on an Nvidia GeForce RTX 3090 24 GB GPU. This resulted on average in $243 \pm 67$ train batches and $104 \pm 29$ test batches. Using transfer learning, we trained each RTRBM for 10,000 epochs with a learning rate of $10^{-3}$. For each epoch we calculated the gradients for 20 batches in parallel before updating the model parameters. This strategy reduced the training

time to a relatively short duration of 3 hr for 10,000 epochs calculated on an *NVIDIA RTX3090* GPU. To maintain neural assemblies we enforced an L1-norm sparsity regulator with its constant set to $\lambda = 10^{-6}$ (with $\lambda = 10^{-7}$ for two of the animals), and a reduced learning rate for the matrix $W$ by two orders of magnitude (i.e. $10^{-5}$).

## Performance evaluation and significance testing

The performance of both the cRBM and RTRBM models after training was evaluated through inspection of the learning moments, similar to that of the simulated data (see see Performance metric for more details). Furthermore, to test the significance of the difference in performance between the RTRBM and the cRBM, we made use of a bootstrap method. To this end, we randomly selected 1000 neurons ($\sim 2\%$ of the total population) and calculated the neural statistics for each model. This process was repeated $n$ times with non-overlapping subsets such that each neuron only gets sampled once. For each repetition, the Spearman correlation between data and model sampled moments was calculated for each $\langle f_k \rangle_{model}$ (see *Figure 2*). We then calculated a confidence interval given by two standard deviations from the mean for the Spearman correlation. If the confidence intervals for the RTRBM and the cRBM did not overlap, the difference in performance was considered significant.

## Clustering of hidden-hidden unit weights

To identify neural assemblies with similar temporal connectivity, agglomerative clustering is employed on the matrix $\hat{U}$ after model training. To this end, we utilize the function *AgglomerativeClustering* from *scikit-learn* and use Ward's method to evaluate the distance dendrogram (*Pedregosa et al., 2011*). Clustering can be applied to the incoming (row) or outgoing (column) connections of the matrix $\hat{U}$, in this work we apply it to the outgoing connections only. For fish 4, as shown in *Figure 3D and E*, a distance threshold of 20 was used to threshold the resulting distance dendrogram, which identified six functional clusters. Receptive fields were determined by assigning neurons to the identified clusters based on the strength of their weight (*Figure 3E*). Here, a strong weight is determined by proportional thresholding, $w_{i,j} > w_{thr}$. Here $w_{thr}$ is set so that at least 5000 neurons have a strong connection towards the hidden layer. This mirrors the thresholding strategy used in *Figure 3B*.

## Alignment of the estimated temporal weight matrix

In the context of the simulated data, the activity of the visible units are systematically generated in assemblies of size $N_v^S$. Nevertheless, during the training process of the RTRBM, which solely relies on the state of the visible units, the link between each assembly and a particular hidden unit becomes arbitrary. This is, of course, under the assumption that all assemblies are retrieved correctly during the training process. Consequently, the estimated temporal weights $\hat{U}$ can have any arbitrary ordering and is often not matched with that of the original matrix $U$ used to generate the data, generally resulting in an invalid match between both matrices. To enable a valid comparison, the ordering of the hidden units in the estimated matrix $\hat{U}$ is matched to those in the original matrix $U$ by leveraging the learned visible-to-hidden unit weight matrix $\hat{W}$. Here, as the assemblies of the visible units are known, assemblies are matched with hidden units under the assumption that an assembly of visible units is most linked to the hidden unit with the largest mean absolute visible-to-hidden weights between them (*Figure 2—figure supplement 1A*, left), calculated using the row-wise correlations with the ideal ('diagonal') weight matrix $W$. The ordering of the hidden units can then be set such that it is aligned with the ordering of the assemblies of visible units. Cases where two assemblies have the same initially matched hidden unit are resolved by sequentially assigning assemblies by order of their correlation strength to the hidden unit. These cases, however, generally indicate that the assemblies of visible units were not correctly captured by the model.

In addition to any arbitrary ordering of hidden units in the estimated matrix $\hat{U}$, it is possible for assemblies to form an inverse but correct match with a hidden unit. This means that the corresponding hidden unit is inactive when the corresponding assembly is active and vice versa. This phenomenon is enabled through the anti-symmetry of the activation function, and generally occurs when the weights between the assembly and the hidden unit are negative. An inverse match is identified based on the sign of the mean weight between the hidden unit and the matched assembly of visible units. In the case of an inverse match with a hidden unit, the temporal weights between this hidden unit and all the other units are inverted, resulting in sign switches in the estimated matrix $\hat{U}$ (*Figure 2—figure*

*supplement 1*, right). In the case of two inverted hidden units, the two temporal weights between them are unaffected as the sign switches cancel each other out.

Together, these corrections yield an aligned estimated temporal weight matrix $\hat{U}$ that is more similar to the original matrix $U$, allowing element-wise comparisons (*Figure 2—figure supplement 1B*). All presented $\hat{U}$ matrices for simulated data are aligned according to this procedure.

### Performance metrics

We evaluated model performance using multiple metrics throughout this study. More specifically, we compared between first and second order model and data statistics for both the visible and hidden units and we used the MSE on the inferred states of the visible units.

### Comparison of model and data statistics

The cRBM and RTRBM models are trained to optimize multiple statics. These include the mean activity of the visible units (neurons) $\langle v_i \rangle$, the mean activity of the hidden units $\langle h_\mu \rangle$ and their paired interaction $\langle v_i h_\mu \rangle$. Furthermore, one can evaluate the second-order statistics $\langle v_i v_j \rangle$, and $\langle h_\mu h_\nu \rangle$ that are not directly optimized by the models.

To evaluate model performance, we take these statistics and compare then between those of the empirical data $\langle f_k \rangle_{data}$ and the model $\langle f_k \rangle_{model}$. Data statistics are calculated based on a withheld test-data set consisting of ~30% of the data. To evaluate hidden-unit statistics for the empirical data, we use the expected value of $h_t$ conditioned on the visible unit state $v_t$ at time point $t$ for the model under evaluation. Model statistics are not explicitly available and must be sampled from the model under study. To this end, model statistics $\langle f_k \rangle_{model}$ are approximated through Gibbs sampling of the visible and hidden unit states. For the simulated data, 15 steps of Gibbs sampling are used with a burn-in period of 4000 steps. This is done for 100 time points of data with a chain length of 20 time-steps. For the zebrafish data, again 15 steps of Gibbs sampling are used with a burn-in period of 4000 time-steps. Here 68–150 random time points in the test set were chosen each with a sampling chain of 14–16 time-steps. The number of time points and chain length depended on the length of the functional recording of a single data-batch, and was empirically chosen by comparing resulting performance.

In all cases, we subsequently measure correspondence between pairs of data and model statistics $\langle f_k \rangle_{data}$ and $\langle f_k \rangle_{model}$ by evaluating their Spearman correlation.

### Predictive quality

The performance of the model in predicting the activity of visible units ahead in time is assessed through a measure of the mean squared error (MSE) between reconstructed data and a test data set. The MSE is defined as

$$\text{MSE}(t) = \frac{1}{N_v} \sum_{i=1}^{N_v} \left( v_i(t) - \hat{v}_i(t) \right)^2 \tag{15}$$

where $N_v$ denotes the number of visible units, and $\hat{v}_i(t)$ is the estimated state of visible unit $i$ at time-step $t$. When predicting $T$ time-steps ahead, $\hat{v}_i(t)$ implicitly depends on $v_i(t-T)$ as discussed above. As the visible units only take on binary states, the MSE is equal to the mean absolute error (MAE) and is also equal to 1 minus the accuracy.

For interpretability, the MSE is normalized (nMSE) such that $\text{nMSE} = 1$ corresponds to a naive unbiased estimator and 0 to an optimal estimator when considering the MSE arising from inherent stochasticity.

$$\text{nMSE} = \frac{\text{MSE} - \text{MSE}_{var}}{\text{MSE}_{naive} - \text{MSE}_{var}} \tag{16}$$

With $\text{MSE}_{naive}$ and $\text{MSE}_{var}$ explained below.

### Determining error bounds for simulated data

For the simulated data, the model predictive performance would be directly interpretable if the model generating the data was completely deterministic. However, the model used in this work features

multiple stochastic components. To gain insight into model performance for simulated data, it is thus necessary to infer statistically meaningful bounds on performance with respect to the generating model. To this end, we here describe methods used for estimating both a lower- and an upper-bound for the generating model used in this work.

### Lower-bound estimation

As described in the methods above, the state of the visible units is sampled through a Poisson distribution from the assembly activity consisting of (1) an intrinsic, time-varying firing rate, generated by randomly timed, Gaussian-shaped variations of firing rate, and (2) recurrent, delayed interactions of firing rates between the hidden populations with interaction time $\Delta t_A$. It is straightforward to estimate the variance in a Poisson sampling process under a known distribution of assembly traces. However, the intrinsic firing rates of the artificial neural assemblies are complicated to describe with theoretical distributions. The exact calculation of this theoretical bound is thus rather complex. Furthermore, one must make a distinction between an estimator with perfect knowledge of the assembly state at the previous time-step, an *ideal estimator*, and a *real estimator* that must approximate this assembly state, such as the RTRBM. A representative lower-bound thus includes this source of stochasticity.

To obtain a lower-bound that includes these notions, we turn to empirical methods. To this end, the assembly activity in a previous time-step is initialized to a fixed, known state. Then, the temporal interactions are calculated to obtain the deterministic assembly activity after an interval $\Delta t_A$ time-steps, which are added to different samples from the intrinsic assembly activity to obtain the full assembly state. Poisson sampling is performed on two such states and the MSE is calculated between them. This corresponds to the performance of an ideal model in predicting the state of the visible units ahead by a time equivalent to the interaction delay, with perfect knowledge of the underlying dynamics, and yields sets of MSE samples incorporating all aforementioned sources of stochasticity.

For a representative distribution of the possible states, 10,000 known assembly activity states are randomly sampled from the original test data, which form the fixed previous time-step. Next, $2 \times 200$ random instances of the intrinsic assembly activities are added as described above per initial state, and samples are taken twice from each of the 200 pairs of states. As in our case $\mathrm{MSE} = \mathrm{MAE}$ is a linear metric, the mean MSE is taken over all initial states and all random intrinsic states, which is our estimate $\mathrm{MSE_{var}}$ The uncertainty in this estimate is quantified as the standard error of the mean, and is calculated by seeing each average MSE per initial state as an estimate of the mean.

### Upper-bound estimation

An upper-bound is determined through a representative naive estimator. Here, we define this naive estimator with the expected firing rate of each visible unit $P(\hat{v}_i^t = 1) = \langle v_i \rangle$, such that the estimator is unbiased by definition. The MSE reached by this estimator can be precisely calculated as $\mathrm{MSE_{naive}} = \langle 2 \langle v_i \rangle \left(1 - \langle v_i \rangle\right) \rangle$. The choice of this unbiased estimator is especially important in the case of sparse activity traces as dealt with here. It is easy to see that a lower MSE is reached by the (biased) estimator $P(\hat{v}_i^t) = 0$ with $\mathrm{MSE} = \langle v_i \rangle$. It is worth noting that in the context of down-sampling experiments, even though all models are assessed using the same test dataset, the act of down-sampling yields a similar but different mean activity $\langle v_i \rangle$, resulting in a slightly different theoretical estimate of the MSE of the naive unbiased estimator.

## Acknowledgements

We would like to thank T van de Plas and G Tubiana for technical discussions. BE acknowledges funding from an NWO VIDI grant (016.Vidi.189.052) and a DFG Forschungsstipendium (EN919/1-1).

## Additional information

### Funding

| Funder | Grant reference number | Author |
|---|---|---|
| Nederlandse Organisatie voor Wetenschappelijk Onderzoek | 016.Vidi.189.052 | Bernhard Englitz |
| Deutsche Forschungsgemeinschaft | EN919/1-1 | Bernhard Englitz |

The funders had no role in study design, data collection and interpretation, or the decision to submit the work for publication.

### Author contributions

Sebastian Quiroz Monnens, Casper Peters, Kasper Smeets, Software, Formal analysis, Investigation, Visualization, Methodology, Writing – original draft, Writing – review and editing; Luuk Willem Hesselink, Supervision, Investigation, Visualization, Methodology, Writing – original draft, Writing – review and editing; Bernhard Englitz, Supervision, Writing – original draft, Project administration, Writing – review and editing

### Author ORCIDs

Luuk Willem Hesselink (iD) https://orcid.org/0009-0003-0735-9913
Kasper Smeets (iD) https://orcid.org/0009-0008-0621-3118
Bernhard Englitz (iD) https://orcid.org/0000-0001-9106-0356

Reviewer #1 (Public review): https://doi.org/10.7554/eLife.98489.3.sa1
Reviewer #2 (Public review): https://doi.org/10.7554/eLife.98489.3.sa2
Author response https://doi.org/10.7554/eLife.98489.3.sa3

---

## Additional files

### Supplementary files

• MDAR checklist

### Data availability

The RTRBM models described in this study have been implemented in Python 3.7. Packages used for model implementation, analysis, and data visualization include Numpy (*Harris et al., 2020*), Scipy (*Virtanen et al., 2020*), Pytorch (*Paszke et al., 2019*), Scikit-learn (*Pedregosa et al., 2011*), matplotlib (*Hunter, 2007*), Pandas (*McKinney, 2010*), Seaborn (*Waskom et al., 2017*), h5py (*Collette, 2013*). Separate notebooks are available allowing for recreation of each individual figure presented in this paper. All code accompanying this paper is available on GitHub (copy archived at *Englitz, 2024*). All accompanying data is hosted on OSF. The model-generated simulated data is seeded and can easily be generated through the accompanying notebooks. All functional whole-brain datasets were obtained in the lab of G Debrégeas and are publicly available here. Copies of the zebrafish datasets used in this project can additionally be found at the OSF data share accompanying this project.

The following dataset was generated:

| Author(s) | Year | Dataset title | Dataset URL | Database and Identifier |
|---|---|---|---|---|
| Hesselink LW, Englitz B, Monnens SLQ, Casper P | 2024 | Zebrafish RTRBM | https://doi.org/10.17605/OSF.IO/TX2VZ | Open Science Framework, 10.17605/OSF.IO/TX2VZ |

The following previously published dataset was used:

| Author(s) | Year | Dataset title | Dataset URL | Database and Identifier |
|-----------|------|---------------|-------------|-------------------------|
| van der Plas TL, Tubiana J, Goc GL, Migault G, Kunst M, Baier H, Bormuth V, Englitz B, Debrégeas G | 2023 | Data repository for Van der Plas, Tubiana and colleagues | https://gin.g-node.org/vdplasthijs/cRBM_zebrafish_spontaneous_data | G-Node, cRBM_zebrafish_spontaneous_data |

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
