## [Editor Report · eLife Assessment]

This study introduces a **useful** extension to a recently proposed model of neural assembly activity. The extension was to add recurrent connections to the hidden units of the Restricted Boltzmann Machine. The authors show **solid** evidence that the new model outperforms their earlier model on both a simulated dataset and on whole-brain neural activity from zebrafish.

---

## [Referee Report · Reviewer #1 (Public review)]

Summary:

Understanding large-scale neural activity remains a formidable challenge in neuroscience. While several methods have been proposed to discover the assemblies from such large-scale recordings, most of previous studies do not explicit modeling the temporal dynamics. This study is an attempt to uncover the temporal dynamics of assemblies using a tool that have been establish in other domains.

The authors previously introduced the compositional Restricted Boltzmann Machine (cRBM) to identify neuron assemblies in zebrafish brain activity. Building upon this, they now employ the Recurrent Temporal Restricted Boltzmann Machine (RTRBM) to elucidate the temporal dynamics within these assemblies. By introducing recurrent connections between hidden units, RTRBM could retrieve neural assemblies and their temporal dynamics from simulated and zebrafish brain data.

Strengths:

The RTRBM has been previously used in other domains. Training the model has been already established. This study is an application of such model to neuroscience. Overall, the paper is well-structured and the methodology is robust, the analysis is solid to support the authors claim.

Weaknesses:

The overall degree of advance is very limited. The performance improvement by RTRBM compared to their cRBM is marginal, and insights into assembly dynamics are limited.

(1) The biological insights from this method are constrained. Though the aim is to unravel neural ensemble dynamics, the paper lacks in-depth discussion on how this method enhances our understanding of zebrafish neural dynamics. For example, the dynamics of assemblies can be analyzed using various tools such as dimensionality reduction methods once we have identified them using cRBM. What information can we gain by knowing the effective recurrent connection between them? It would be more convincing to show this in real data.

(2) Including predicted and measured neural activity traces could aid readers in evaluating model efficacy. The current version only contains comparison of the statistics, such as mean and covariance.

---

## [Referee Report · Reviewer #2 (Public review)]

Summary:

In this work, the authors propose an extension to some of the last author's previous work, where a compositional restricted Boltzmann machine was considered as a generative model of neuron-assembly interaction. They augment this model by recurrent connections between the Boltzmann machine's hidden units, which allow them to explicitly account for temporal dynamics of the assembly activity. Since their model formulation does not allow the training towards a compositional phase (as in the previous model), they employ a transfer learning approach according to which they initialise their model with a weight matrix that was pre-trained using the earlier model so as to essentially start the actually training in a compositional phase. Finally, they test this model on synthetic and actual data of whole-brain light-sheet-microscopy recordings of spontaneous activity from the brain of larval zebrafish.

Strengths:

This work introduces a new model for neural assembly activity. Importantly, being able to capture temporal assembly dynamics is an interesting feature that goes beyond many existing models. While this work clearly focuses on the method (or the model) itself, it opens up an avenue for experimental research where it will be interesting to see if one can obtain any biologically meaningful insights considering these temporal dynamics when one is able to, for instance, relate them to development or behaviour.

Weaknesses:

For most of the work, the authors present their RTRBM model as an improvement over the earlier cRBM model. Yet, when considering synthetic data, they actually seem to compare with a "standard" RBM model. This seems odd considering the overall narrative and that when considering whole-brain zebrafish data, the comparisons were made between RTRBM and cRBM models. For that, the RTRBM model was initialised with the cRBM weight matrix to overcome the fact that RTRBM alone does not seem to converge to a compositional phase, so to cite the latter as reason does not really make sense.

Furthermore, whether the clusters shown in Figure 3E can indeed be described as "spatially localized" is debatable. Especially in view of clusters 3 and 4, this seems a stretch. If receptive fields are described as "spatially localized", arguably, one would expect that they are contained in some small (compared to the overall size of the brain) or specific anatomical brain region. However, this is clearly not the case here.

In addition, the performance comparison for the temporal dynamics of the hidden units actually suggests that the RTRBM (significantly) underperforms where the text says (Line 235f) it outperforms the cRBM model.

---

## [Author Response]

The following is the authors’ response to the original reviews.

**Reviewer #1 (Public Review):**
Summary:Understanding large-scale neural activity remains a formidable challenge in neuroscience. While several methods have been proposed to discover the assemblies from such large-scale recordings, most previous studies do not explicitly model the temporal dynamics. This study is an attempt to uncover the temporal dynamics of assemblies using a tool that has been established in other domains.The authors previously introduced the compositional Restricted Boltzmann Machine (cRBM) to identify neuron assemblies in zebrafish brain activity. Building upon this, they now employ the Recurrent Temporal Restricted Boltzmann Machine (RTRBM) to elucidate the temporal dynamics within these assemblies. By introducing recurrent connections between hidden units, RTRBM could retrieve neural assemblies and their temporal dynamics from simulated and zebrafish brain data.Strengths:The RTRBM has been previously used in other domains. Training in the model has been already established. This study is an application of such a model to neuroscience. Overall, the paper is well-structured and the methodology is robust, the analysis is solid to support the authors' claim.Weaknesses:The overall degree of advance is very limited. The performance improvement by RTRBM compared to their cRBM is marginal, and insights into assembly dynamics are limited.(1) The biological insights from this method are constrained. Though the aim is to unravel neural ensemble dynamics, the paper lacks in-depth discussion on how this method enhances our understanding of zebrafish neural dynamics. For example, the dynamics of assemblies can be analyzed using various tools such as dimensionality reduction methods once we have identified them using cRBM. What information can we gain by knowing the effective recurrent connection between them? It would be more convincing to show this in real data.

See below in the recommendations section.

(2) Despite the increased complexity of RTRBM over cRBM, performance improvement is minimal. Accuracy enhancements, less than 1 in synthetic and zebrafish data, are underwhelming (Figure 2G and Figure 4B). Predictive performance evaluation on real neural activity would enhance model assessment. Including predicted and measured neural activity traces could aid readers in evaluating model efficacy.

See below in the recommendations section.

**Recommendations:**
(1) The biological insights from this method are constrained. Though the aim is to unravel neural ensemble dynamics, the paper lacks in-depth discussion on how this method enhances our understanding of zebrafish neural dynamics. For example, the dynamics of assemblies can be analyzed using various tools such as dimensionality reduction methods once we have identified them using cRBM. What information can we gain by knowing the effective recurrent connection between them? It would be more convincing to show this in real data.

We agree with the reviewer that our analysis does not explore the data far enough to reach the level of new biological insights. For practical reasons unrelated to the science, we cannot further explore the data in this direction at this point, however, funding permitting, we will pick up this question at a later stage. The only change we have made to the corresponding figure at the current stage was to adapt the thresholds, which better emphasizes the locality of the resulting clusters.

(2) Despite the increased complexity of RTRBM over cRBM, performance improvement is minimal. Accuracy enhancements, less than 1 in synthetic and zebrafish data, are underwhelming (Figure 2G and Figure 4B). Predictive performance evaluation on real neural activity would enhance model assessment. Including predicted and measured neural activity traces could aid readers in evaluating model efficacy.

We thank the reviewer kindly for the comments on the performance comparison between the two models. We would like to highlight that the small range of accuracy values for the predictive performance is due to both the sparsity and stochasticity of the simulated data, and is not reflective of the actual percentage in performance improvement. To this end, we have opted to use a rescaled metric that we call the normalised Mean Squared Error (nMSE), where the MSE is equal to 1 minus the accuracy, as the visible units take on binary values. This metric is also more in line with the normalised Log-Likelihood (nLLH) metric used in the cRBM paper in terms of interpretability. The figure shows that the RTRBM can significantly predict the state of the visible units in subsequent time-steps, whereas the cRBM captures the correct time-independent statistics but has no predictive power over time.

We also thank the reviewer for pointing out that there is no predictive performance evaluation on the neural data. This has been chosen to be omitted for two reasons. First, it is clear from Fig. 2 that the (c)RBM has no temporal dependencies, meaning that the predictive performance is determined mostly by the average activity of the visible units. If this corresponds well with the actual mean activity per neuron, the nMSE will be around 0. This correspondence is already evaluated in the first panel of 3F. Second, as this is real data, we can not make an estimate of a lower bound on the MSE that is due to neural noise. Because of this, the scale of the predictive performance score will be arbitrary, making it difficult to quantitatively assess the difference in performance between both models.

(3) The interpretation of the hidden real variable rt lacks clarity. Initially interpreted as the expectation of h[t], its interpretation in Eq (8) appears different. Clarification on this link is warranted.

We thank the reviewer kindly for the suggested clarification. However, we think the link between both values should already be sufficiently clear from the text in lines 469-470:

“Importantly, instead of using binary hidden unit states 𝐡[𝑡−1], sampled from the expected real valued hidden states 𝐫[𝑡−1], the RTRBM propagates these real-valued hidden unit states directly.”

In other words, both indeed are the same, one could sample a binary-valued 𝐡[𝑡-1] from the real-valued 𝐫[𝑡-1] through e.g. a Bernoulli distribution, where 𝐫[𝑡-1] would thus indeed act as an expectation over 𝐡[𝑡−1]. However, the RTRBM formulation keeps the real-valued 𝐫[𝑡-1] to propagate the hidden-unit states to the next time-step. The motivation for this choice is further discussed in the original RTRBM paper (Sutskever et al. 2008).

(4) In Figure 3 panel F, the discrepancy in x-axis scales between upper and lower panels requires clarification. Explanation regarding the difference and interpretation guidelines would enhance understanding.

Thank you for pointing out the discrepancy in x-axis scales between the upper and lower panels of Figure 3F. The reason why these scales are different is that the activation functions in the two models differ in their range, and showing them on the same scale would not do justice to this difference. But we agree that this could be unclear for readers. Therefore we added an additional clarification for this discrepancy in line 215:

“While a direct comparison of the hidden unit activations between the cRBM and the RTRBM is hindered by the inherent discrepancy in their activation functions (unbounded and bounded, respectively), the analysis of time-shifted moments reveals a stronger correlation for the RTRBM hidden units (rs=0.92, p<ϵ) compared to the cRBM (rs=0.88, p<ϵ)”

(5) Assessing model performance at various down-sampling rates in zebrafish data analysis would provide insights into model robustness.

We agree that we would have liked to assess this point in real data, to verify that this holds as well in the case of the zebrafish whole-brain data. The main reason why we did not choose to do this in this case is that we would only be able to further downsample the data. Current whole brain data sets are collected at a few Hz (here 4 Hz, only 2 Hz in other datasets), which we consider to be likely slower than the actual interaction speed in neural systems, which is on the order of milliseconds between neurons, and on the order of ~100 ms (~10 Hz) between assemblies. Therefore reducing the rate further, we expect to only see a reduction in quality, which we considered less interesting than finding an optimum. Higher rates of imaging in light-sheet imaging are only achievable currently by imaging only single planes (which defies the goal of whole brain recordings), but may be possible in the future when the limiting factors (focal plane stepping and imaging) are addressed. For completeness, we have now performed the downstepping for the experimental data, which showed the expected decrease in performance. The results have been integrated into Figure 4.

**Reviewer #2 (Public Review):**
Summary:In this work, the authors propose an extension to some of the last author's previous work, where a compositional restricted Boltzmann machine was considered as a generative model of neuron-assembly interaction. They augment this model by recurrent connections between the Boltzmann machine's hidden units, which allow them to explicitly account for temporal dynamics of the assembly activity. Since their model formulation does not allow the training towards a compositional phase (as in the previous model), they employ a transfer learning approach according to which they initialise their model with a weight matrix that was pre-trained using the earlier model so as to essentially start the actually training in a compositional phase. Finally, they test this model on synthetic and actual data of whole-brain light-sheet-microscopy recordings of spontaneous activity from the brain of larval zebrafish.Strengths:This work introduces a new model for neural assembly activity. Importantly, being able to capture temporal assembly dynamics is an interesting feature that goes beyond many existing models. While this work clearly focuses on the method (or the model) itself, it opens up an avenue for experimental research where it will be interesting to see if one can obtain any biologically meaningful insights considering these temporal dynamics when one is able to, for instance, relate them to development or behaviour.Weaknesses:For most of the work, the authors present their RTRBM model as an improvement over the earlier cRBM model. Yet, when considering synthetic data, they actually seem to compare with a "standard" RBM model. This seems odd considering the overall narrative, and it is not clear why they chose to do that. Also, in that case, was the RTRBM model initialised with the cRBM weight matrix?

Thank you for raising the important point regarding the RTRBM comparison in the synthetic data section. Initially, we aimed to compare the performance of the cRBM with the cRTRBM. However, we encountered significant challenges in getting the RTRBM to reach the compositional phase. To ensure a fair and robust comparison, we opted to compare the RBM with the RTRBM.

A few claims made throughout the work are slightly too enthusiastic and not really supported by the data shown. For instance, when the authors refer to the clusters shown in Figure 3D as "spatially localized", this seems like a stretch, specifically in view of clusters 1, 3, and 4.

Thanks for pointing out this inaccuracy. When going back to the data/analyses to address the question about locality, we stumbled upon a minor bug in the implementation of the proportional thresholding, causing the threshold to be too low and therefore too many neurons to be considered.

Fixing this bug reduces the number of neurons, thereby better showing the local structure of the clusters. Furthermore, if one would lower the threshold within the hierarchical clustering, smaller, and more localized, clusters would appear. We deliberately chose to keep this threshold high to not overwhelm the reader with the number of identified clusters. We hope the reviewer agrees with these changes and that the spatial structure in the clusters presented are indeed rather localized.

Moreover, when they describe the predictive performance of their model as "close to optimal" when the down-sampling factor coincided with the interaction time scale, it seems a bit exaggerated given that it was more or less as close to the upper bound as it was to the lower bound.

We thank the reviewer for catching this error. Indeed, the best performing model does not lay very close to the estimated performance of an optimal model. The text has been updated to reflect this.

When discussing the data statistics, the authors quote correlation values in the main text. However, these do not match the correlation values in the figure to which they seem to belong. Now, it seems that in the main text, they consider the Pearson correlation, whereas in the corresponding figure, it is the Spearman correlation. This is very confusing, and it is not really clear as to why the authors chose to do so.

Thank you for identifying the discrepancy between the correlation values mentioned in the text and those presented in the figure. We updated the manuscript to match the correlation coefficient values in the figure with the correct values denoted in the text.

Finally, when discussing the fact that the RTRBM model outperforms the cRBM model, the authors state it does so for different moments and in different numbers of cases (fish). It would be very interesting to know whether these are the same fish or always different fish.

Thank you for pointing this out. Keeping track of the same fish across the different metrics makes sense. We updated the figure to include a color code for each individual fish. As it turns out each time the same fish are significantly better performing.

**Recommendations:**
Figure 1: While the schematic in A and D only shows 11 visible units ("neurons"), the weight matrices and the activity rasters in B and C and E and F suggest that there should be, in fact, 12 visible units. While not essential, I think it would be nice if these numbers would match up.

Thank you for pointing out the inconsistency in the number of visible units depicted in Figure 1. We agree that this could have been confusing for readers. The figure has been updated accordingly. As you suggested, the schematic representation now accurately reflects the presence of 12 visible units in both the RBM and RTRBM models.

Figure 3: Panel G is not referenced in the main text. Yet, I believe it should be somewhere in lines 225ff.

Thank you for mentioning this. We added in line 233 a reference to figure 3 panel G to refer to the performance of the cRBM and RTRBM on the different fish.

Line 637ff: The authors consider moments ⟨vihμ⟩ and ⟨vihj⟩, and from the context, it seems they are not the same. However, it is not clear as to why because, judging from the notation, they should be the same.

The second-order statistic ⟨vihj⟩ on line 639 was indeed already mentioned and denoted as ⟨vihμ⟩ on line 638. It has now been removed accordingly in the updated manuscript.

I found the usage of Û and U throughout the manuscript a bit confusing. As far as I understand, Û is a learned representation of U. However, maybe the authors could make the distinction clearer.

We understand the usage of Û and U throughout the text may be confusing for the reader. However, we would like to notify the reviewer that the distinction between these two variables is explained in line 142: “in addition to providing a close estimate (Û) to the true assembly connectivity matrix U”. However, for added clarification to the reader, we added additional mentions of the estimated nature of Û throughout the text in the updated manuscript.

Equation 3: It would be great if the authors could provide some more explanation of how they arrived at the identities.

These identities have previously been widely described in literature. For this reason, we decided not to include their derivation in our manuscript. However, for completeness, we kindly refer to:

Goodfellow, I., Bengio, Y., & Courville, A. (2016). Chapter 20: Deep generative models [In *Deep Learning*]. MIT Press. https://www.deeplearningbook.org/contents/generative_models.html

Typos:- L. 196: "connectiivty" -> "connectivity"- L. 197: Does it mean to say "very strong stronger"?- L. 339: The reference to Dunn et al. (2016) should appear in parentheses.- L. 504f: The colon should probably be followed by a full sentence.- Eq. 2: In the first line, the potential V still appears, which should probably be changed to show the concrete form (-b * h) as in the second line.- L. 351: Is there maybe a comma missing after "cRBM"?- L. 271: Instead of "correlation", shouldn't it rather be "similarity"? - L. 218: "Figure 3D" -> "Figure 3F"

We thank the reviewer for pointing out these typos, which have all (except one) been fixed in the text. We do emphasize the potential V to show that there are alternative hidden unit potentials that can be chosen. For instance, the cRBM utilizes dReLu hidden unit potentials.

**Reviewer #3 (Public Review):**
With ever-growing datasets, it becomes more challenging to extract useful information from such a large amount of data. For that, developing better dimensionality reduction/clustering methods can be very important to make sense of analyzed data. This is especially true for neuroscience where new experimental advances allow the recording of an unprecedented number of neurons. Here the authors make a step to help with neuronal analyses by proposing a new method to identify groups of neurons with similar activity dynamics. I did not notice any obvious problems with data analyses here, however, the presented manuscript has a few weaknesses:(1) Because this manuscript is written as an extension of previous work by the same authors (van der Plas et al., eLife, 2023), thus to fully understand this paper it is required to read first the previous paper, as authors often refer to their previous work for details. Similarly, to understand the functional significance of identified here neuronal assemblies, it is needed to go to look at the previous paper.

We agree that the present Research Advance has been written in a way that builds on our previous publication. It was our impression that this was the intention of the Research Advance format, as spelled out in its announcement "eLife has introduced an innovative new type of article – the Research Advance – that invites the authors of any eLife paper to present significant additions to their original research". In the previous formatting guidelines from eLife this was more evident with a strong limitation on the number of figures and words, however, also for the present, more liberal guidelines, place an emphasis on the relation to the previous article. We have nonetheless tried in several places to fill in details that might simplify the reading experience.

(2) The problem of discovering clusters in data with temporal dynamics is not unique to neuroscience. Therefore, the authors should also discuss other previously proposed methods and how they compare to the presented here RTRBM method. Similarly, there are other methods using neural networks for discovering clusters (assemblies) (e.g. t-SNE: van der Maaten & Hinton 2008, Hippocluster: Chalmers et al. 2023, etc), which should be discussed to give better background information for the readers.

The clustering methods suggested by the reviewer do not include modeling any time dependence, which is the crucial advance presented here by the introduction of the RTRBM, in extending the (c)RBM. In our previous publication on the cRBM (an der Plas et al., eLife, 2023), this comparison was part of the discussion, although it focussed on a different set of methods. While clustering methods like t-SNE, UMAP and others certainly have their value in scientific analysis, we think it might be misleading the reader to think that they achieve the same task as an RTRBM, which adds the crucial dimension of temporal dependence.

(3) The above point to better describe other methods is especially important because the performance of the presented here method is not that much better than previous work. For example, RTRBM outperforms the cRBM only on ~4 out of 8 fish datasets. Moreover, as the authors nicely described in the Limitations section this method currently can only work on a single time scale and clusters have to be estimated first with the previous cRBM method. Thus, having an overview of other methods which could be used for similar analyses would be helpful.

We think that the perception that the RTRBM performs only slightly better is based on a misinterpretation of the performance measure, which we have tried to address (see comments above) in this rebuttal and the manuscript. In addition we would like to emphasize that the structural estimation (which is still modified by the RTRBM, only seeded by the cRBMs output), as shown in the simulated data, makes improved structural estimates, which is important, even in cases where the performance is comparable (which can be the case if the RBM absorbs temporal dependencies of assemblies into modified structure of assemblies). We have clarified this now in the discussion.

**Recommendations:**
(1) Line 181: it is not explained how a reconstruction error is defined.

Dear reviewer, thanks for pointing this out. A definition of the (mean square) reconstruction error is added in this line.

(2) How was the number of hidden neurons chosen and how does it affect performance?

Thank you for pointing this out. Due to the fact that we use transfer learning, the number of hidden units used for the RTRBM is given by the number of hidden units used for training the cRBM. In further research, when the RTRBM operates in the compositional phase, we can exploit a grid search over a set of hyper parameters to determine the optimal set of hidden units and other parameters.